# In vivo assessment of the neural substrate linked with vocal imitation accuracy

Julie Hamaide[1], Kristina Lukacova[2], Jasmien Orije[1], Georgios A Keliris[1], Marleen Verhoye[1], Annemie Van der Linden[1]*

[1]Bio-Imaging Lab, Department of Biomedical Sciences, University of Antwerp, Wilrijk, Belgium; [2]Centre of Biosciences, Institute of Animal Biochemistry and Genetics, Slovak Academy of Sciences, Bratislava, Slovakia

**Abstract** Human speech and bird song are acoustically complex communication signals that are learned by imitation during a sensitive period early in life. Although the brain areas indispensable for speech and song learning are known, the neural circuits important for enhanced or reduced vocal performance remain unclear. By combining in vivo structural Magnetic Resonance Imaging with song analyses in juvenile male zebra finches during song learning and beyond, we reveal that song imitation accuracy correlates with the structural architecture of four distinct brain areas, none of which pertain to the song control system. Furthermore, the structural properties of a secondary auditory area in the left hemisphere, are capable to predict future song copying accuracy, already at the earliest stages of learning, before initiating vocal practicing. These findings appoint novel brain regions important for song learning outcome and inform that ultimate performance in part depends on factors experienced before vocal practicing.

## Introduction

Human speech and bird song are highly complex and rapid motor behaviours that are learned by imitation from adults and serve to produce complex communication signals vital for social interactions (*Brainard and Doupe, 2013*). Both are acquired during a temporally well-defined sensitive period early in life which consists of a sensory learning phase where a perceptual target of speech sounds or a song model are memorised, followed by a sensorimotor learning phase, where human infants or juvenile birds engage in intense vocal practicing and will try to match the spectro-temporal characteristics of their own vocalisations to the previously established perceptual targets based on multi-sensory feedback (*Doupe and Kuhl, 1999*; *Tchernichovski et al., 2001*). Zebra finches, an established model to study certain aspects of human speech acquisition (*Brainard and Doupe, 2013*), learn to sing a single song that crystallises (i.e. remains unchanged) for life after the sensorimotor learning phase. The success of the song learning process can be accurately quantified by computing the acoustic similarity between the song learned and sang by the pupil and the original tutor song that is used as a reference (*Tchernichovski et al., 2000*). Intriguingly, only zebra finch males learn to sing. As a result, juvenile female zebra finches do not experience a sensorimotor and crystallisation phase for song learning like male zebra finches do. However, female zebra finches do rely on early exposure to song to form an auditory memory of tutor song to develop song preference (*Hauber et al., 2013*; *Bolhuis and Moorman, 2015*; *Chen et al., 2016*).

In songbirds, sensory learning is thought to depend on a distributed set of areas including several auditory regions (*Bolhuis and Gahr, 2006*; *Hahnloser and Kotowicz, 2010*). The motor aspect of song is encoded by cortical areas (*Wild, 1997*), which receive input from the vocal basal ganglia to introduce variability in vocal motor output during sensorimotor practicing (*Bottjer et al., 1984*). Auditory feedback signals that encode errors in own performance were detected in the secondary auditory areas (*Keller and Hahnloser, 2009*) and are likely transmitted to the dopaminergic

*For correspondence:
annemie.vanderlinden@
uantwerpen.be

Competing interests: The authors declare that no competing interests exist.

midbrain nuclei (*Mandelblat-Cerf et al., 2014*), via the ventral pallidum (*Chen et al., 2019*). The dopaminergic midbrain nuclei and ventral pallidum were recently found to steer vocalisations towards the desired vocal target during sensorimotor song learning (*Hisey et al., 2018*; *Chen et al., 2019*). Each of these pathways is indispensable for sensorimotor learning to succeed, and during the sensorimotor learning phase, the precise interplay of these systems is tightly set. Furthermore, the behavioural difference between male and female zebra finches (only males sing) is reflected in the structural organisation of the zebra finch brain. That is, the song control system, a circuitry composed of well-delineated brain areas that are interconnected by fibre pathways and that are responsible for singing behaviour, is more enhanced in males compared to females (*MacDougall-Shackleton and Ball, 1999*). These structural differences include disparities in local volume, interconnectivity as well as intrinsic microstructural tissue properties (e.g. soma size of LMAN) (*Nixdorf-Bergweiler, 1998*). The differences between male and female birds in volume of the song control system nuclei arise during the song learning period (when juvenile males actively engage in vocal motor practicing [*Nixdorf-Bergweiler, 1996*].

Importantly, however, much like human speech, bird song is a complex culturally transmitted socially learned communication skill. Indeed, several studies suggest that social factors other than merely auditory access to a singing tutor (e.g. via auditory playback) are important for vocal learning. For example, live tutoring results in higher song similarity scores compared to what is achieved by tape tutored birds (*Derégnaucourt et al., 2013*). Social feedback that juveniles receive from the adult male and female zebra finch (its caretakers) during the sensory and sensorimotor period of vocal learning is capable of influencing song learning accuracy levels (*Chen et al., 2016*; *Carouso-Peck and Goldstein, 2019*). Therefore, it is expected that additional brain areas and neural circuits outside the traditionally studied auditory and song control system may be involved and perhaps are capable of influencing ultimate song performance (i.e. learning accuracy) levels.

These foregoing studies that aimed at unravelling which neural circuits are indispensable for song learning to succeed, used highly targeted methods such as permanent lesioning, temporarily inactivating or optogenetic neuromodulation of specific brain regions during for example tutoring sessions. Despite the exquisite spatial and/or temporal resolution of the latter research tools, they require strong a priori hypotheses of regions expected to be involved and findings obtained with such methods inform on the implication of (only) these specific brain regions (neural populations) in the process under investigation. In contrast, alternative research tools exist that enable to repeatedly and over extended time frames capture the structural and/or functional properties of the entire brain when subjects advance from the basic to advanced performance levels. Combining this brain-wide information with data-driven processing methods enable an unbiased identification of neural correlates that would otherwise be missed when using a hypothesis-driven region-of-interest (ROI) research strategy. Over the last decades, Magnetic Resonance Imaging (MRI) in combination with voxel-based analyses emerged as a prominent tool to uncover, follow and quantify plastic changes of brain function and structure arising along training and learning paradigms in humans and small animal species (*Dayan and Cohen, 2011*; *Sagi et al., 2012*; *Zatorre et al., 2012*). Despite the correlational nature of such findings, they benefit from a significant advantage as they allow to establish spatio-temporal maps that indicate when –along a training or learning paradigm– and where in the brain specific neuroplastic events occur and, as such, provide a strong basis for further in depth testing using highly sensitive and specific research methods that are capable of understanding the precise functional implications of the previously identified targets.

Inspired by such imaging studies, we set up a longitudinal study in both male and female zebra finches where we repeatedly collected structural MRI data of the entire zebra finch brain at six time points during and one time point after the critical period for song learning (*Figure 1A*). Using voxel-based statistical testing, we established spatio-temporal maps that revealed that (1) sex differences in local tissue microstructure exclusively co-localise with the song control nuclei and arise along the song learning process, and (2) most myelin-containing brain structures exhibit structural maturational changes between 20 and 200 dph. Further, starting from the advanced sensorimotor learning phase when the male pupils have already mastered most of the tutor song, and proceeding up to the stage where they reach full song crystallisation, we recorded the song sung by the male pupils the first days after the MRI session and computed the song learning accuracy level of the pupils by comparing the spectro-temporal properties of the pupil song to those of the tutor song. Exploiting the advantages of in vivo MRI, we performed brain-wide voxel-based correlational analyses to explore

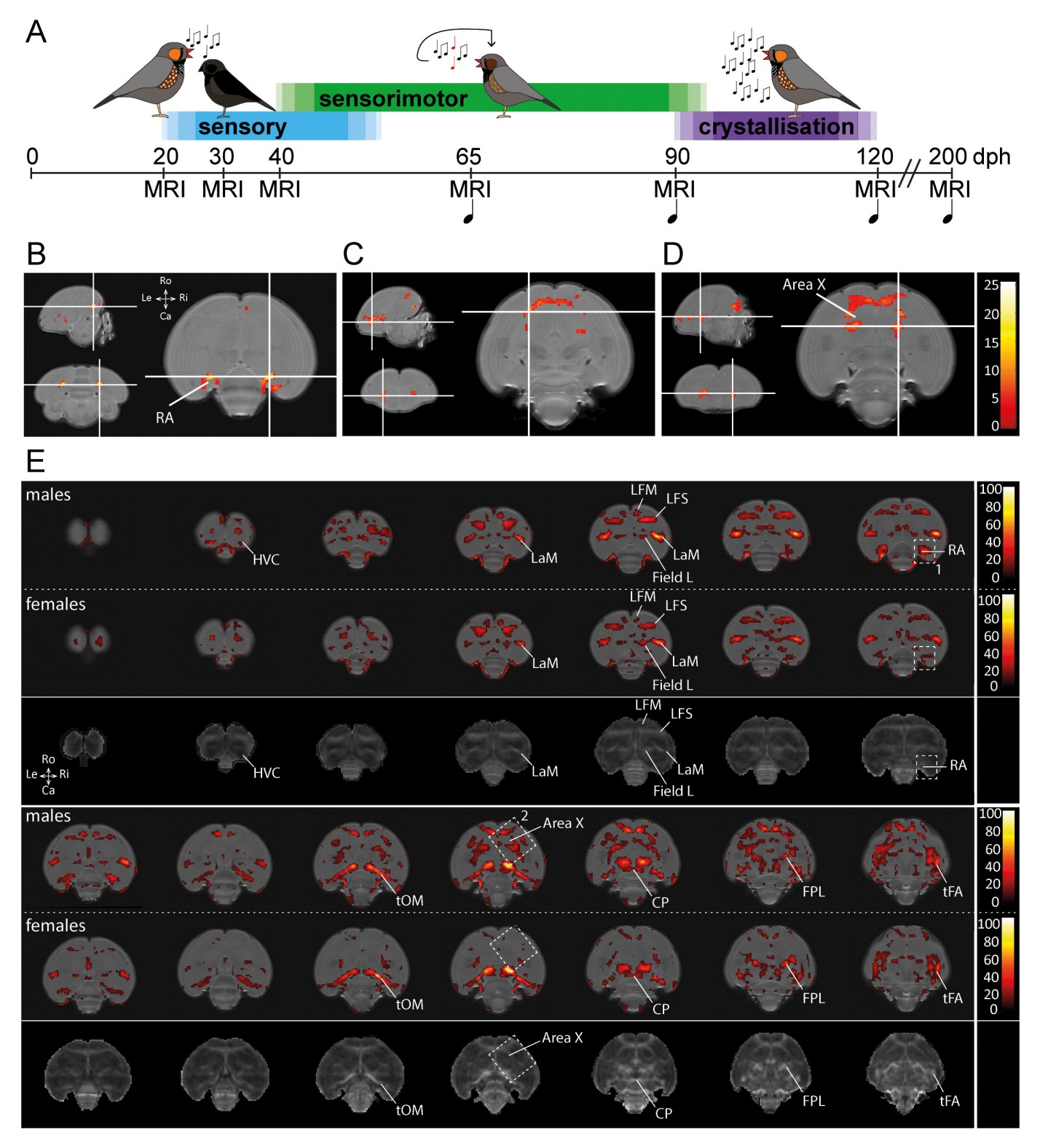

**Figure 1.** Monitoring the maturing zebra finch brain in male and female finches. (A) Study design. Structural MRI data were acquired during the sensory phase (20, 30 dph), sensorimotor phase (40, 65 dph), crystallisation phase (90, 120 dph) and well after song is mastered (200 dph) in male zebra finches. Structural MRI data were obtained at the same time points in female zebra finches. During each imaging session, we collected a DTI and 3D anatomical scan. The first days after each imaging session, we recorded the song sung by the juvenile male birds ( ♪ ), starting from 65 dph (only male zebra finches sing). (B-D) Statistical parametric maps highlighting voxels that display an interaction between age*sex for Fractional Anisotropy (FA), one of the

*Figure 1 continued on next page*

*Figure 1 continued*

parameters of DTI, during ontogeny (**B-D**). The crosshairs converge at the arcopallium (**B**), rostro-lateral Area X surroundings (**C**), and caudal Area X surroundings (**D**). The maps are thresholded at $p_{uncorrected}$ <0.001; $k_E \geq 30$ voxels (**B-D**), and overlaid on the population-based template. The statistical maps are colour-coded according to the scales on the right. (**E**) Statistical brain maps illustrating the main effect of age of FA for male and female zebra finches. The results are displayed in accordance with pFWE <0.001 kE $\geq 5$ voxels, and overlaid on the population-based template. The statistical maps are colour-coded according to the scales on the right. FA values range from 0 (black) to 1 (white). The areas demarcated by white-dotted lines refer to clusters identified in the interaction between age and sex with 1 (RA) and 2 (Area X) (**B-D**). The third and sixth rows present an average FA map calculated from FA maps of male birds obtained at 200 dph and serves to identify the anatomical (mostly white matter) structures covered by a significant cluster. See *Supplementary file 1* for statistics. Abbreviations: LaM: mesopallial lamina; LFM: supreme frontal lamina; LFS: superior frontal lamina; tOM: occipitomesencephalic tract; CP: posterior commissure; FPL: lateral prosencephalic fascicle; tFA: fronto-arcopallial tract; Do: dorsal; Ve: ventral; Ro: rostral; Ca: caudal.

for relationships between song learning accuracy and local tissue volume (3D structural imaging) or the intrinsic tissue properties (Diffusion Tensor Imaging (DTI)) along the different stages of the song learning process. Even though the strength of DTI mainly lies in its sensitivity to detect changes in white matter, many studies have demonstrated its contribution in identifying microstructural learning-related changes in grey matter areas such as for example the hippocampus in humans and rats (*Sagi et al., 2012*) (for review *Hamaide et al., 2016*). Likewise, we discovered that the structural properties of specific parts of the secondary auditory cortices, that is left caudomedial nidopallium (NCM) and the caudomedial mesopallium (CM), and unexpectedly, the ventral pallidum (VP) and the fronto-arcopallial tract (tFA) correlate with song imitation accuracy, while the different nuclei of the SCS seemed not involved. Indepth analyses revealed that the structural properties of left NCM and the tFA mainly reflect relationships between performance and structure at the population-level, while the structural architecture of the CM and VP appears to change along the sensorimotor learning process, in individual birds. Fascinatingly, we also discovered that the structural properties of left NCM are predictive of future learning accuracy already before pupils actively engage in vocal practicing during the early phases of sensory learning.

## Results

### Longitudinal structural MRI changes in the brains of maturing male and female zebra finches

We set up a longitudinal study where we repeatedly collected structural MRI data of the entire zebra finch brain (*Figure 1A*). These data consist of 3D anatomical scans and DTI scans. The 3D anatomical dataset enabled us to assess regional changes in brain volume that arise over time (brain development) or between male and female zebra finch brains (sex differences; these data have been published *Hamaide et al., 2017*). The DTI datasets allow to establish spatiotemporal maps that indicate when and where in the brain neuroplastic changes in tissue microstructure occur (*Hamaide et al., 2016*). In the current study, we focus on the Fractional Anisotropy (FA) outcome, a metric derived from DTI data. FA quantifies the directional dependence of water diffusion and hence indirectly reflects specific microstructural tissue characteristics (*Beaulieu, 2002*). Note that alterations in FA can be caused by a wide variety of microstructural tissue re-organisations including altered axonal integrity, myelination, axon diameter and density, change in cellular morphology, etc. (*Beaulieu, 2002*; *Zatorre et al., 2012*; *Dyrby et al., 2018*). Using non-invasive in vivo structural MRI, we have been able to detect sex differences in local volume in the maturing zebra finch brain (*Hamaide et al., 2018a*) and in intrinsic microstructural tissue properties in adult birds (*Hamaide et al., 2017*). The present data enables us to extend on the latter, as the present study includes DTI data obtained in juvenile zebra finches.

To unveil specific brain structures that display a sexual dimorph developmental trajectory of DTI properties that matches with the sexual dimorph song production behaviour, we tested for an interaction between age and sex using a voxel-based repeated measures ANOVA on the smoothed DTI parameter maps. Only those clusters that survive $p$FWE <0.05 and kE $\geq 5$ voxels were considered significant. Such an interaction was detected in clusters bilateral at the Area X surroundings and the arcopallium (*Figure 1B–D*).

Since several brain areas displayed an interaction between age and sex over time, the main effect of age was explored in male and female birds separately. The data illustrate that most myelin containing structures and lamina of the juvenile zebra finch brain mature between 20 and 200 dph (*Figure 1E*). These lamina carry diverse collections of axonal fibres connecting distinct brain areas. Both in male and female birds, the clusters displaying a significant difference in FA covers almost the entire path of the occipitomesencephalic tract (tOM), from the arcopallium passing the rostral border of the thalamus before traveling ventrally towards the diencephalon. Furthermore, parts of the superior frontal lamina (LFS), mesopallial lamina (LaM), the caudo-dorsal extension of the Area X surroundings, anterior commissure, lateral prosencephalic fascicle (FPL) and fronto-arcopallial tract (tFA) could be observed as well. Interestingly, in females (less in males) a small portion of Field L showed a change as well.

In addition, several clusters identified by the (voxel-based) interaction between age and sex over time (*Figure 1B*) were only found to be significantly changing during ontogeny in males (indicated by the white dotted boxes in *Figure 1E*).

## Song performance improves even after crystallisation

We extracted the acoustical properties of individual song syllables at each age of the male zebra finches to evaluate how the spectral and temporal structure of the syllables evolve from (advanced) plastic to fully mature stereotyped and crystallised song. The duration of the inter-syllable intervals gradually shortens from sensorimotor to song crystallisation and even after song crystallisation towards 200 dph ($p < 0.0001$ $F_{(3, 38.2)} = 13.8789$; *Supplementary file 2*). This indicates that birds gradually sing faster. Syllable Wiener entropy scores decrease during the sensorimotor phase ($p = 0.0032$ $F_{(3, 37.8)} = 5.48$; *Figure 2—figure supplement 1*), meaning that the syllables gain in tonality. None of the pitch-related measures, or frequency and amplitude modulation changed over time (*Supplementary file 2*). Further, by quantifying the standard deviation of the spectro-temporal features, we observe that syllables are sung with lower acoustic variability when song crystallises between 90 and 120 dph (*Figure 2—figure supplement 1*). These analyses informed that the syllables gain tonality (lower acoustic noisiness), become more structured (less noisy) and less acoustically diverse along sensorimotor song maturation. Furthermore, while the spectral content of syllables mainly forms during the sensorimotor phase, the temporal properties of the songs continue to change beyond the crystallisation phase. This corroborates previous studies in zebra finches (*Glaze and Troyer, 2013*).

To quantitatively assess fine-scale changes in song performance and to estimate song learning accuracy, we evaluated the progression of song learning from the advanced sensorimotor phase (65 days post hatching (dph)), over the crystallisation phase (90–120 dph), to fully crystallised song (200 dph; *Figure 1A*). We quantified how successful the juvenile birds (pupils) learned, that is copied, the tutor song, by computing the spectral similarity between the pupil and the tutor song (*Tchernichovski et al., 2000*). Song similarity to tutor song increased gradually from 65 to 200 dph ($p = 0.0251$ $F_{(3, 37.0)} = 3.4890$; *Figure 2A*) reaching similar levels as described by others (*Tchernichovski et al., 2000*; *Derégnaucourt and Gahr, 2013*). Moreover, as song crystallisation results in a highly stereotyped, consistent order of syllables within a motif, sequence stereotypy also increased gradually from 65 to 200 dph ($p = 0.0052$ $F_{(3, 38.4)} = 4.7904$, *Figure 2B*).

Notably, the overall variability in song similarity between birds was quite large indicating that not all birds copy the tutor song equally well. Prior studies have shown that successful song learning not only depends on the ability of the juveniles to hear the song model: social interactions between the juvenile and adult birds are crucial mediators in determining the overall quality of the song copy (*Chen et al., 2016*; *Carouso-Peck and Goldstein, 2019*). Consistent with this, we observed that song similarity depended on tutor identity, that is, juveniles will consistently sing a good or less good tutor song copy depending on their tutor ($p = 0.0159$ $F_{(7, 6.1)} = 6.7597$; *Figure 2A* – colours reflect tutor identity). This main effect of tutoring bird on song similarity outcome did not seem to depend on the length of the songs, as based on similarity scores obtained at 200 dph, no differences in the number of syllables for birds having a 'high' (>68%) or 'low' (<68%) similarity score could be observed (*Supplementary file 3*).

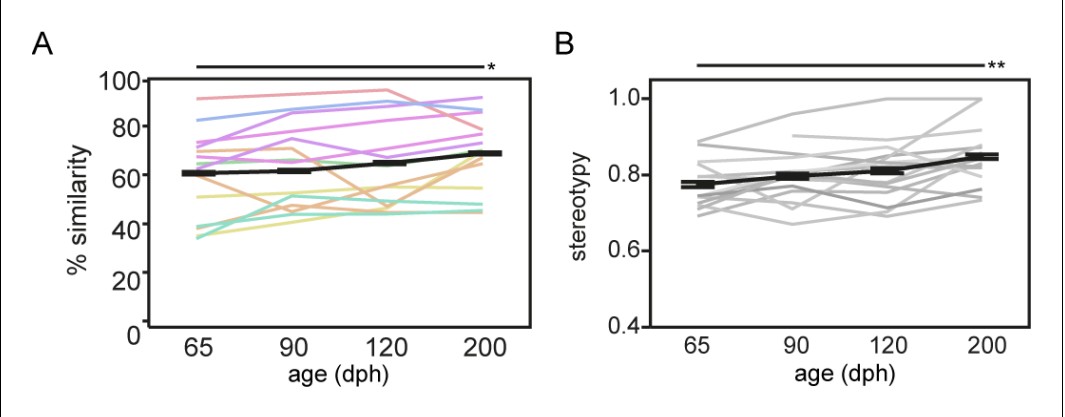

**Figure 2.** Song similarity improves beyond crystallisation. Graph **A** and **B** refer to respectively song similarity to tutor song and song stereotypy in function of age. Both increase from 65 to 200 dph (mixed-effect model main effect of age: song similarity: p=0.0251 $F_{(3, 37.0)}$=3.4890; sequence stereotypy: p=0.0052 $F_{(3, 38.4)}$=4.7904). Each thin coloured or grey line refers to the average performance of an individual bird over the different ages. The bold black line presents the average group performance (mean ± SEM; n = 14; 20 data points per time point per bird). The colour-code of the lines in A encodes tutor identity, that is birds raised by the same tutor share the same colour (see *Supplementary file 12*). The colour-code illustrates that song similarity is dependent on tutor identity (mixed-effect model main effect for tutor: p=0.0159 $F_{(7, 6.1)}$=6.7597). Asterisks indicate significant differences over time identified by a mixed model analysis with post hoc Tukey's HSD test. *: p<0.05; **p<0.01. Abbreviations: dph: days post-hatching.

The online version of this article includes the following figure supplement(s) for figure 2:

**Figure supplement 1.** Song syllable scores.

## Song learning accuracy traces back to the CM, VP, tFA and NCM in the sensorimotor learning phase

Even though song performance improves from the sensorimotor to the crystallisation phase (and even beyond), not all birds learn to reproduce the tutor song equally well. Therefore, we set out to explore whether better song performance (i.e. more accurate tutor song copying), correlates with a specific structural signature in the brain. Inspired by ample in vivo imaging studies describing training- or learning-induced brain-behaviour relationships (*Zatorre et al., 2012*), we performed brain-wide voxel-based statistical analyses to highlight potential brain sites that present a correlation between song learning accuracy (% similarity between pupil and tutor song) and local volume (log-transformed modulated jacobian determinant (log mwj)) or intrinsic tissue properties derived from the DTI metrics, that is Fractional Anisotropy (FA) (*Beaulieu, 2002*; *Zatorre et al., 2012*). These brain-wide voxel-based analyses uncovered four clusters (*Figures 3–4* and *Supplementary file 4*). Using various atlases of the zebra finch brain (http://www.zebrafinchatlas.org/; *Nixdorf and Bischof, 2007*; *Poirier et al., 2008*; *Karten et al., 2013*) and high-resolution tract tracings within the zebra finch brain (*Hamaide et al., 2017*), we identified that these clusters co-localise with two secondary auditory areas, that is the caudomedial nidopallium (NCM) and caudal mesopallium (CM), with a white matter tract that connects the basorostral nucleus to the arcopallium (frontoarcopallial tract (tFA) *Wild and Farabaugh, 1996*), and with an area at the base of the telencephalon termed the ventral pallidum (VP). The VP has a direct role in sensorimotor song learning (*Chen et al., 2019*) and it contains many fibres of passage that project from the midbrain dopaminergic nuclei to Area X of the anterior forebrain pathway and Area X-DLM projections that pertain to the song control system (*Gale et al., 2008*).

Correlations between song similarity and FA were observed in the left tFA (peak: $p_{FWE}$ <0.001 T = 6.81; *Figure 3A*), and in left NCM (rostral NCM; peak: $p_{FWE}$ = 0.019 T = 5.69; *Figure 3E* and *Figure 3—figure supplement 1*). Furthermore, we found an additional cluster midsagittal near the striatum and mesopallium, extending laterally and caudo-ventrally adjacent to the septomesencephalic tract (TSM; sub-peak next to the TSM in the left hemisphere: $p_{FWE}$ = 0.002 T = 6.38;

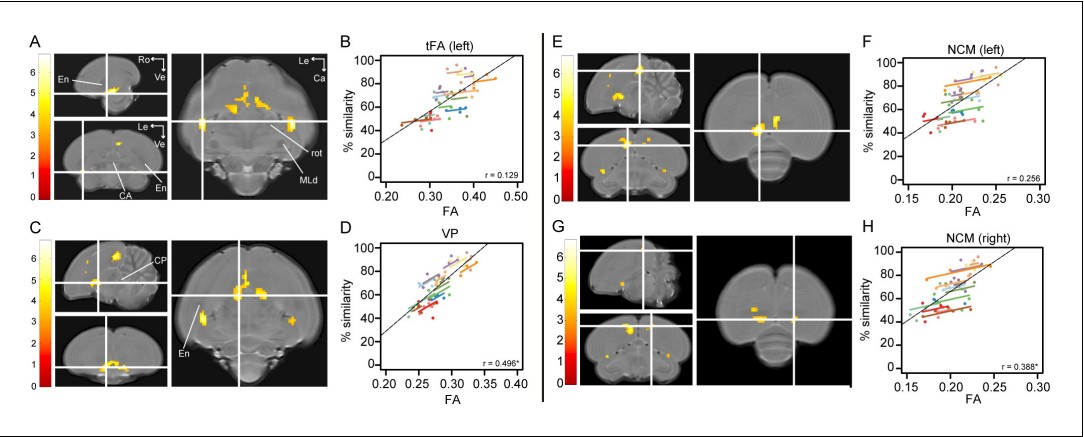

**Figure 3.** Song imitation accuracy correlates positively with Fractional Anisotropy (FA) in the tFA (**A–B**), VP (**C–D**) and NCM (**E–H**). The statistical maps (**A, C, E, G**) present the outcome of the voxel-based multiple regression testing for a correlation between song similarity and FA (n = 14). The crosshairs point to the tFA in the left hemisphere (**A**), the VP (**C**), and NCM in the left (**E**) and right (**G**) hemisphere. Results are overlaid on the population-based MRI template and scaled according to the colour-code (T values) on the left of each statistical map. Only voxels that reached $p_{uncorrected}$ <0.001 and take part of a cluster of at least 40 contiguous voxels are displayed. Graphs **B, D, F and H** visualise the nature of the correlation between song similarity and FA where individual data points are colour-coded according to bird-identity (i.e. one colour = one bird). The average within-bird correlation is presented by the coloured lines, while the black dashed line indicates the overall association between song similarity and FA, disregarding bird-identity or age. 'r' is the repeated-measures correlation (rmcorr) coefficient. The * indicates a significant rmcorr correlation between FA and % similarity in the VP (p=0.001) and right NCM (p=0.0121). Abbreviations: CA: anterior commissure; CP: posterior commissure; En: entopallium; MLd: dorsal part of the lateral mesencephalic nucleus; rot: nucleus rotundus; tFA: fronto-arcopallial tract; VP: ventral pallidum; Le: left; Ca: caudal; Ro: rostral; Ve ventral. See *Supplementary file 4* for statistics.

The online version of this article includes the following figure supplement(s) for figure 3:

**Figure supplement 1.** Song imitation accuracy correlates positively with fractional anisotropy (FA) in NCM.

*Figure 3C*). Based on this spatial pattern and in accordance with the Karten-Mitra zebra finch brain atlas (http://www.zebrafinchatlas.org/; *Karten et al., 2013*), we identified this area as the VP. Interestingly, when inspecting the statistical maps at a less conservative threshold ($p_{uncorrected}$ <0.001 $k_E$ ≥40 voxels), clusters could be also observed at the right tFA (peak: $p_{FWE}$ = 0.001 $T$ = 6.42) and the right NCM (peak: $p_{FWE}$ = 0.032 $T$ = 5.55; *Figure 3G*). Also at this less conservative threshold, we observe that the cluster covering the left NCM extends rostro-laterally towards the CM (sub-peak of NCM cluster: $p_{FWE}$ = 0.194 $T$ = 5.01; *Figure 3G* and *Figure 3—figure supplement 1*).

To test if learning accuracy correlates with the local volume of specific brain areas (log mwj), we performed voxel-based analysis and found a significant anticorrelation in two brain areas, that is the VP (peak: $p_{FWE}$ <0.001 $T$ = 8.06; *Figure 4A*) and the medial and lateral portions of the CM rostral to field L (resp. CMM and CLM) potentially including nucleus avalanche (Av; left: peak: $p_{FWE}$ = 0.001 $T$ = 7.10; right: peak: $p_{FWE}$ <0.001 $T$ = 7.42; *Figure 4C*).

## Learning-related relationships versus between-bird variance

The voxel-based correlation analysis detects an overall association between song performance (% similarity) and the structural properties of the brain without taking the repeated measures into account. As a result, these analyses cannot infer whether the brain-behaviour associations are mainly driven by between-subject variation in performance and structure, or whether individual improvements in song imitation relate to specific structural properties of the clusters at specific learning periods. To make this distinction, we first extracted for each bird and each time point separately the mean log mwj or mean FA from the voxel-based clusters. Next, we performed (i) Spearman's correlation analysis (ρ) to characterise potential correlations between the structural properties of the cluster-based ROIs and song similarity at 65 or 200 dph, and (ii) a repeated-measures correlation

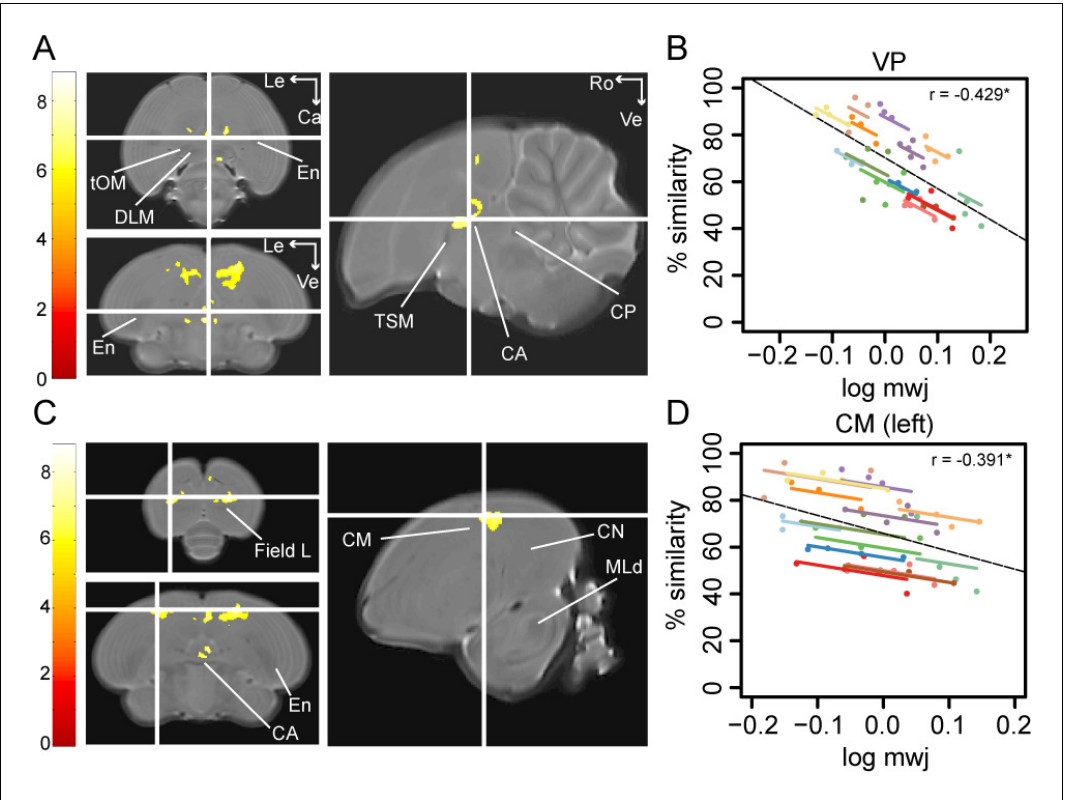

**Figure 4.** Song imitation accuracy correlates negatively with the local volume of the VP (**A–B**) and the CM (**C–D**). The statistical parametric maps present the outcome of the voxel-based multiple regression testing for a correlation between song similarity and local tissue volume (n = 14) and are visualised at $p_{FWE}$ <0.05 and $k_E \geq 80$ voxels, and overlaid on the population-based template. The crosshairs point to the VP (**A**) or the CM in the left hemisphere (**C**). T-values are colour-coded according to the scale immediately left to the SPMs. Graphs **B** and **D** inform on the nature of the association between song similarity (%) and log-transformed modulated jacobian determinant (log mwj; a metric reflecting local tissue volume). More specifically, the individual data points of the graphs are colour-coded according to bird-identity (i.e. one colour = one bird). The average within-bird correlation is presented by the coloured lines, while the dashed black line indicates the overall association between song similarity and log mwj, disregarding bird-identity or age. 'r' is the repeated-measures correlation (rmcorr) coefficient. The * indicates a significant rmcorr correlation between logmwj and % similarity in the VP (p=0.0057) and left CM (p=0.0126). Abbreviations: CA: anterior commissure; CM: caudal mesopallium; CN: caudal nidopallium; CP: posterior commissure; DLM: medial part of the dorsolateral nucleus of the anterior thalamus; En: entopallium; MLd: dorsal part of the lateral mesencephalic nucleus; tOM: occipitomesencephalic tract; TSM: septo-mesencephalic tract; VP: ventral pallidum; Le: left; Ca: caudal; Ve: ventral; Ro: rostral. See *Supplementary file 5* for statistics.

analysis (rmcorr *Bakdash and Marusich, 2017*; *Figure 3B,D,F,H*; *Figure 4B,D*) which takes for each bird the repeated-measures into account and can provide inference on the common association between brain structure and song similarity across the group of birds. In summary, this analysis can provide inference on potential learning-related changes in local brain structure. The outcome of the Spearman's correlation and rmcorr analyses, including Benjamini-Hochberg correction for multiple comparisons, are summarised in *Table 1* and *Supplementary file 6*, *7*.

In two of the clusters identified above, that is the NCM and the tFA, the association between song similarity and FA appeared to be driven by between-subject variance (65 dph: NCM: left: p=0.0081; right: p=0.0138; tFA: left: p=0.0009; right: p=0.0045; 200 dph: NCM: left: p=0.0070; right: p=0.0336; tFA: left: p=0.0012; right: p=0.0041). Surprisingly, a small cluster in the right NCM displayed, in addition, a significant repeated-measures correlation. This indicates that when individual birds improved their performance, FA increased accordingly in the right NCM.

**Table 1.** Summary of within- and between-subject correlations of the cluster-based ROIs.

| Correlation between | Cluster-based ROI | Hemisphere | Rmcorr | | 65 dph | | 200 dph | |
|---|---|---|---|---|---|---|---|---|
| | | | R | P | Spearmans' ρ | P | Spearmans' ρ | P |
| % similarity and FA | tFA | Left | 0.1290 | 0.4200 | **0.7846** | **0.0009** | **0.7714** | **0.0012** |
| | | Right | 0.0215 | 0.8940 | **0.7099** | **0.0045** | **0.7143** | **0.0041** |
| | NCM | Left | 0.2560 | 0.1060 | **0.6747** | **0.0081** | **0.6835** | **0.0070** |
| | | Right | **0.3880** | **0.0121** | **0.6396** | **0.0138** | **0.5692** | **0.0336** |
| | VP | | **0.4960** | **0.0010** | **0.8154** | **0.0004** | **0.7890** | **0.0008** |
| % similarity and log mwj | VP | | **−0.4290** | **0.0057** | **−0.7978** | **0.0006** | −0.4418 | 0.1138 |
| | CM | Left | **−0.3910** | **0.0126** | −0.5297 | 0.0514 | −0.3055 | 0.2882 |
| | | Right | **−0.4160** | **0.0075** | −0.4769 | 0.0846 | −0.1912 | 0.5126 |

'log mwj' refers to the log-transformed, modulated and warped jacobian determinants; FA stands for Fractional Anisotropy, one of the DTI metrics. 'r' is the repeated-measures correlation coefficient of the within-subject correlation analyses. Spearmans' ρ informs on potential correlations between the MRI parameters and song similarity at a specific time point between birds. Tests that survive Benjamini-Hochberg FDR correction for multiple comparisons are highlighted in bold (**Supplementary file 6**, **7**) Abbreviations: dph: days post hatching.

In contrast, the CM presented no between-subject correlations at any age. This suggests that birds that sing a better copy of the tutor song do not typically exhibit a smaller or larger volume of this specific part of the CM. However, individual improvements in song learning resulted in a lower local volume of the CM (left: p=0.0126 rmcorr = −0.391; right: p=0.0075 rmcorr = −0.416; *Figure 4D*). The VP, on the other hand, presented significant repeated-measures correlations between song similarity and local volume or FA (log mwj: p=0.0057 rmcorr = −0.429; FA: p=0.0010 rmcorr = 0.496; *Figure 4B*). This suggests that when individual birds learn to produce a more accurate copy of the tutor song, the VP scales down (low mwj) and obtains a more structured organisation (FA). Furthermore, local volume and FA correlate significantly with song similarity in the VP during the sensorimotor phase at 65 dph (log mwj: p=0.0006 Spearman ρ = −0.7978; FA: p=0.0004 Spearman ρ = 0.8154); however, only the correlation between song similarity and FA is maintained until after song crystallisation (p=0.0008 Spearman ρ = 0.7890). The results of the correlation analyses are summarised in *Table 1* and *Supplementary file 6*, *7*.

Together, these findings suggest that birds that sing a better copy of the tutor song have higher FA values in NCM, the VP and the tFA, both during the sensorimotor phase as well as after song crystallisation. Furthermore, learning-related individual advances in producing a more accurate acoustic copy of the tutor song correlate with local tissue structure in the caudal mesopallium and VP. Overall, higher song similarity is related to a smaller volume of the VP and CM or a higher FA (more ordered structure) in the VP, tFA and NCM. Higher FA might refer to a more accurate alignment of fibres or increased myelination in the VP and tFA (*Beaulieu, 2002*), while in grey matter-like structures such as NCM, higher FA values might allude to changes in cell morphology (spines and dendrite branching), alignment or density, etc. (*Zatorre et al., 2012*).

## Song learning accuracy does not trace back to the song control system

The song control system is known to be crucial during the song learning process and its constituents are known to change in volume and in microstructural tissue properties during the first four months of post-hatch life (*Figure 1E* and *Nixdorf-Bergweiler (1996)* Journal of Comparative Neurology). Ample studies have shown that lesioning components of this circuitry during the song learning phase will prevent the pupils from copying the tutor song (*Scharff and Nottebohm, 1991*; *Brainard and Doupe, 2001*). Intriguingly, none of the clusters observed in the current study, when correlating between song performance and brain structure, overlapped with any component of the song control system. Previous studies of the authors, however, have shown that the MRI methods used in this study are sensitive enough to detect correlations between song performance and brain structure in the song control system nuclei for example HVC (*Hamaide et al., 2018b*; *Orije et al., 2020*). To better understand this lack of clusters co-localised with the song control system nuclei in this study, we delineated the song control nuclei based on DTI maps and extracted the structural properties (MRI

parameters) of these regions-of-interest, that is HVC, Area X, LMAN and RA. Then, we tested for correlations between the structural readouts and song similarity. No correlations could be observed (*Figure 5*). Even when including the regions which changed significantly during ontogeny in male zebra finches, that is surrounding Area X and RA (*Figure 1B,D*) in the correlation analysis, no association between brain structure and song similarity could be identified (*Figure 5C,F*). This shows that although the song control system might than be responsible for enabling song learning, the ultimate song performance level (% similarity with tutor song) is determined by a complex sets of circuits that synapse onto the song control system.

## Microstructural characteristics of NCM as defined by FA can predict future good or bad learning outcome even before pupils engage in vocal practicing in the sensory learning phase

The Spearman correlation analyses uncovered that FA values in the VP, NCM and tFA present a clear between-subject correlation with song learning accuracy. This suggests that in the sensorimotor phase good or bad learners are characterised by a distinct structural MRI parameter readout in these regions. Next, based on the scans of the same birds acquired during the sensory (20, 30 dph) and early sensorimotor (40 dph) phases, we evaluated whether similar signs of future good or bad song copying outcome would already be visible in the structural properties of these regions in the early sensorimotor phase, or even before sensorimotor practicing during the sensory learning phase when birds memorise the tutor song but are not yet fully engaged in trial-and-error vocal practicing (*Brainard and Doupe, 2013*).

To this end, we divided the group of male birds into 'good' and 'bad' learners based on the overall song performance obtained at 65–200 dph. More specifically, good learners (n = 7) always sung acoustically accurate copies (>65–68% song similarity to tutor song), while bad learners (n = 5) never produced a copy better than 65–68% similarity to tutor song. Birds that traversed the 65–68%

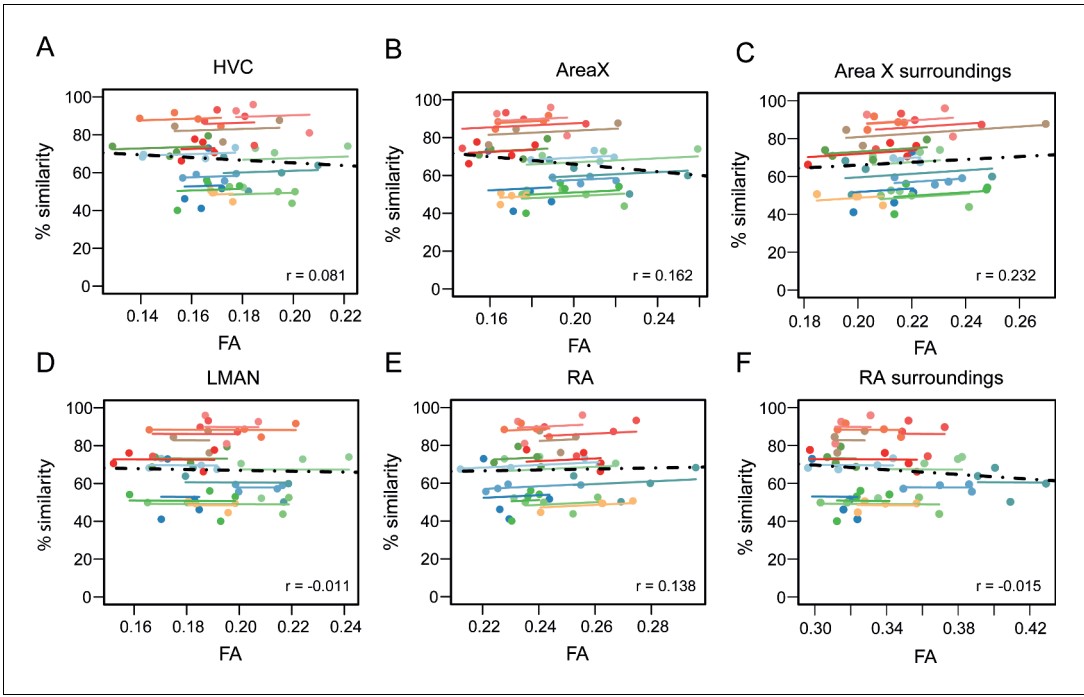

**Figure 5.** Repeated measures correlation showing no correlation between % song similarity and Fractional Anisotropy (FA) of song control system regions. HVC (A), Area X (B), LMAN (D) and RA (E) and the surroundings of song control system nuclei Area X (C) and RA (F) as defined by the clusters derived from interaction age*sex shown in *Figure 1B,D*. The individual data points of the graphs are colour-coded according to bird-identity (i.e. one colour = one bird). The average within-bird correlation is presented by the coloured lines, while the dashed black line indicates the overall association between song similarity and FA, disregarding bird-identity or age. 'r' is the repeated-measures correlation (rmcorr) coefficient.

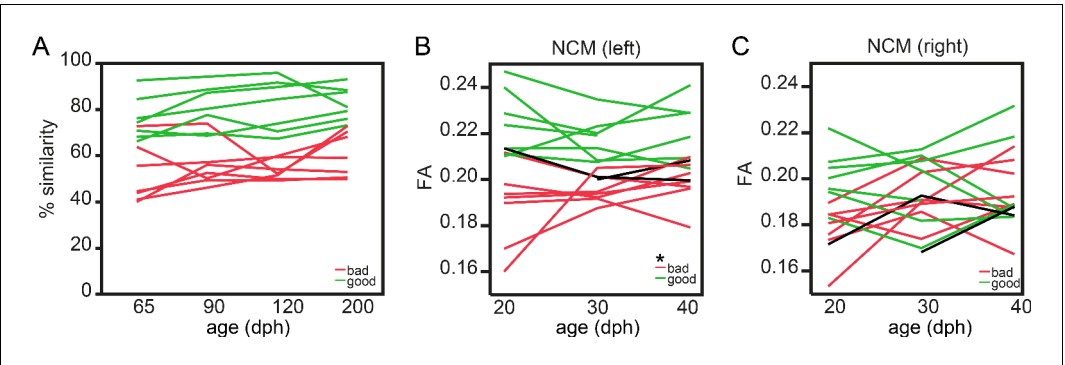

**Figure 6.** Fractional anisotropy in left NCM predicts future song learning accuracy. Graph **A** presents the learning curve of the good (green; n = 7) and bad (red; n = 7) learning birds from 65 to 200 dph. Details on the distinction between good and bad learners can be found in the Results section and in *Supplementary file 2*. Graphs B and C present the difference of FA in NCM between good (green) and bad (red) vocal learners during the sensory (20–30 dph) and early sensorimotor (40 dph) phase in, respectively, the left (**B**) and right (**C**) hemisphere. Each line represents repeated measures obtained from one bird. The * indicates a significant main effect of future learning accuracy (good versus bad learners) for FA in left NCM (mixed model: p<0.0001 $F_{(1,12.3)}$=39.2690).
The online version of this article includes the following figure supplement(s) for figure 6:

**Figure supplement 1.** The structural properties of the VP, CM and tFA cannot predict future song learning accuracy.
**Figure supplement 2.** Fractional anisotropy in left NCM predicts future song learning accuracy.

interval throughout the study (n = 2) were assigned to the 'bad learners' group as at the end of the song learning phase they reached similarity scores < 68% (*Figure 6A*). Next, we tested for an interaction between age (20-30-40 dph) and future learning accuracy (good, bad) in the cluster-based ROIs (*Figure 6—figure supplement 1*). None of cluster-based ROIs survived FDR correction for multiple comparisons when testing for an interaction between age and future learning accuracy. Interestingly, FA in the left NCM displayed a significant main effect of learning accuracy (good versus bad) already at the ages 20–40 dph (p<0.0001 $F_{(1, 12.3)}$=39.2690; *Figure 6B*). Furthermore, FA in left NCM at 20 dph was positively correlated (p=0.01, ρ = 0.662) to the % song similarity at 200 dph (*Figure 6—figure supplement 2*). To our surprise, future good learners consistently demonstrated higher FA values in left NCM compared to bad learners already at the earliest stages of the sensory learning phase. Thus, this structural signature is already present much before the juvenile finches engage in vocal practicing, when the pupils have been exposed to their tutor for several days (*Immelmann, 1969*). This result was specific to the left NCM as none of the other regions identified in the voxel-based correlation analysis showed a similar predictive relationship (*Figure 6C*, *Figure 6—figure supplements 1* and *2*).

## Discussion

We employed hypothesis-free data-driven brain-wide in vivo structural MRI tools and used song similarity to tutor song as a proxy for vocal learning accuracy, starting from the advanced sensorimotor phase up to post-crystallisation song refinement. This unbiased approach led to the observation that the structural properties of the secondary auditory cortices, that is left NCM, the CM, the VP, and – unexpectedly– the tFA correlate with imitation accuracy. Furthermore, between- and within- subject correlation analyses revealed that the structural properties of left NCM and the tFA are mainly caused by between-subject variation in performance and structure, while the structural architecture of the CM and VP appears to change along the sensorimotor learning process, in individual birds. Importantly, we also demonstrated that the structural properties of left NCM during the sensory phase (i.e. when birds establish a memory of the tutor song but have not yet initiated sensorimotor vocal practicing) were different in birds that in the future would become good or bad song learners, as such predicting this quality. Overall, the present findings (i) add a new dimension to previously

published data as we provide clear evidence of relationships between performance levels and the structural properties of four specific areas, (ii) identify a novel not-yet-explored brain area (tFA) in the context of song learning which deserves in-depth investigation in future studies and (iii) uncover that future performance levels can be predicted based on the structural properties of a secondary auditory region at the earliest stages of song learning.

The observation that the structural properties of NCM are predictive of future song copying accuracy already at 20 dph is consistent with a previously established functional role of NCM in establishing a memory of the tutor song during the sensory phase (*Bolhuis and Gahr, 2006*; *London and Clayton, 2008*; *Hahnloser and Kotowicz, 2010*; *Ahmadiantehrani and London, 2017*). Furthermore, tutor-song evoked immediate early gene (IEG) expression levels appear stronger in the left compared to the right NCM and song similarity correlates with the extent of lateralised IEG expression (*Moorman et al., 2012*). Intriguingly, our observation that mainly the left rather than the right NCM presents a correlation between tissue microstructure and tutor song imitation accuracy (*Figure 3E–H* and *Figure 6B–C*) is reminiscent of leftward lateralisation of speech and language processing in humans (*Bishop, 2013*), and is consistent with a recently discovered asymmetry in tissue microstructure in the planum temporale (related to auditory speech processing) in human subjects (*Ocklenburg et al., 2018*), likewise measured by diffusion MRI.

It is noteworthy that the quality of the song learning process (similarity between pupil and tutor song) can only be assessed when juvenile birds engage in vocal practicing and thus it is always a product of both sensory and sensorimotor learning. Benefiting from the non-invasive nature of MRI, however, we were able to opt for a longitudinal study design where we followed birds for a total period of 6 months, from very early on, that is before birds engage in vocal practicing at 20 dph, up to 200 dph when they sing a fully crystallised song. As such, we were able 'to go back in time' and to relate song learning accuracy levels obtained at 65 dph (and older) to the structural properties of specific brain regions of the same birds at 20 dph. Importantly however, we should note that the study design employed in this study does not allow distinguishing between the implications of innate learning bias (innate properties of the pupil) and social enhancers of the tutor (social enhancers that promote learning in pupils). Such tests require a carefully balanced/controlled study design where genetic brothers are raised in different conditions: (1) by its biological father (tutor = bio father), (2) by foster fathers (one foster father (tutor) per genetic brother). Given that the behaviour of the tutor and social interactions between the tutor and the juvenile males have been shown to be important influencers of the juveniles' song learning performance (*Chen et al., 2016*), the rearing conditions and tutor exposure should be carefully controlled for. A yoked experimental design similar to what was used by *Chen et al. (2016)* could help understand the effect of social interaction versus innate learning bias (auditory experience with limited visual and physical interactions). Furthermore, a recent study has found that the interactions between juvenile males and their (foster) mother also have important effects on the juvenile males' song maturation/performance (*Carouso-Peck and Goldstein, 2019*). Therefore, additional conditions that control for effects of learning by social enhancement of adult females should be incorporated as well. Lastly, delaying tutor exposure to after the first measurement (e.g. first (MRI) measure at 30 dph and introduction to tutor at 31 dph), can help differentiate between innate learning bias and social enhancement of vocal learning.

It is generally known that in normal rearing conditions song learning accuracy improves with age. We also clearly observed this effect of learning over time in the current study (*Figure 2*). The current study design does not allow to perfectly quantify to what extent the correlations observed in the voxel-based clusters are driven by age or by general brain developmental processes. However, based on several observations, we speculate that the relationships observed in this study (*Figures 3–4*) are mainly reflecting brain-behaviour relationships related to learning accuracy (the latter of which also improves with age) rather than age-related brain changes (that accidently connect to improvements in learning). Firstly, a previous study by our lab investigated brain development in (juvenile) zebra finches (*Hamaide et al., 2018a*). This study clearly shows that most changes in brain volume occur relatively early (before 65 dph), and that the changes affect large portions of the brains. Furthermore, the same study shows that relatively large, widespread brain areas decrease in volume from 65 dph to 200 dph (the same time frame as this study). The clusters detected in the current study are much smaller and may perhaps overlap with only a small fraction of these large clusters. If

the correlations would mainly be driven by 'aging' or brain development, we would expect a similar profile of clusters as those that were found in the overall effect of time (*Figure 1B*) and in the brain development study (*Hamaide et al., 2018a*). Secondly, *Figure 2B* clearly indicates that not all birds follow a similar learning curve. That is, for some birds, song similarity increases more between 65 and 90 dph, while other pupils show the steepest increase in performance level between 90 and 200 dph. Moreover, not all birds reach similar levels of song learning accuracy. This observation indicates that there is an important source between-subject variance and that age or time only cannot suffi-cient explain performance levels. Based on these observations, we performed Spearman correlation analyses based on the cluster-based ROI data obtained selectively at one specific age, that is 65 dph or 200 dph, to ensure that no time-related effects (brain development) could obscure the outcome. These analyses clearly informed that between-bird variance in song learning accuracy drive the corre-lations detected by the voxel-based analyses. Thirdly, based on literature, it is known that the song control system nuclei change in volume with age (Nixdorf-Bergweiler, 1006 Journal of Comparative Neurology) and are involved in the song learning process. As clearly stated above, we do not find any relationship between song learning accuracy and the structural properties of the brain in the song control system or its immediate surroundings. Based on these observations, we conclude that even though we cannot fully remove age-effects, we strongly believe that the current findings are mainly driven by correlations between performance levels and the structural characteristics of the brain rather than purely brain development effects.

While the structural properties of left NCM already appear to differentiate between future good and bad performing birds in the sensory phase, the local volume and/or intrinsic tissue properties of the VP and tFA are not different between birds with a differential future learning outcome. Further-more, the local volume of the CM and VP decreases when birds progressively become better at sing-ing the tutor song. These findings suggest that the structural properties of the CM, VP and tFA change along the sensorimotor phase. Sensorimotor song refinement requires mechanisms that link motor commands with the associated sensory feedback such that it enables to detect and correct errors in own performance (*Murphy et al., 2017*). Interestingly, recent evidence appoints an impor-tant role to respectively the secondary auditory area CM, the VP and dopaminergic midbrain nuclei that synapse onto the vocal basal ganglia in sensorimotor learning (*Keller and Hahnloser, 2009*; *Hisey et al., 2018*; *Chen et al., 2019*). More specifically, the CM is reciprocally connected with NCM (*Vates et al., 1996*) and presents clear song-selective responses (*Bauer et al., 2008*). A spe-cific sub-field within the CM, the nucleus Avalanche (Av) exhibits singing-driven IEG expression that correlates with the amount of singing in normally hearing and in deafened birds (*Jarvis and Notte-bohm, 1997*). Av-projecting HVC neurons convey premotor signals to the Av and genetic ablation of HVC$_{Av}$ neurons after sensory learning significantly impairs sensorimotor learning in juvenile male zebra finches (*Roberts et al., 2017*). In sum, its song selective neural responses (*Bauer et al., 2008*), its reciprocal connectivity with NCM (*Vates et al., 1996*), and premotor input from HVC (*Roberts et al., 2017*) that might participate in the generation of error-detection signals (*Keller and Hahnloser, 2009*), set the CM as prime target capable of comparing own song performance towards pre-set performance goals such as the memorised tutor song. Our findings complement these previ-ous reports by showing that besides the functional properties, also the structural features of the CM change at par with performance throughout the sensorimotor learning phase.

*Gale et al. (2008)* identified that the VP carries (i) neuronal projections from the arcopallium to dopaminergic midbrain nuclei –a pathway important in error-detection and correction mechanisms necessary for sensorimotor learning (*Mandelblat-Cerf et al., 2014*)– and (ii) axonal collaterals of the DLM-projecting Area X cells –that make up the basal ganglia-thalamic component of the song con-trol system– project to the VP where they synapse onto VTA/SNc projecting neurons (*Gale and Per-kel, 2010*). Using optogenetic neuromodulation, a recent study found mechanistic evidence that these dopaminergic Area X projecting VTA neurons are indispensable for sensorimotor song learn-ing in ontogeny (*Hisey et al., 2018*). Furthermore, VP neurons signal performance error during sing-ing and lesioning VP in juvenile male zebra finches significantly impairs song learning (*Chen et al., 2019*). Besides carrying dopaminergic projections, the VP contains cholinergic neurons as well. Neu-rons located in the VP send cholinergic projections to two cortical regions responsible for the motor aspects of singing, that is HVC and RA (*Li and Sakaguchi, 1997*), and are capable of suppressing HVCs' neural responses to birds' own song by manipulation of the cholinergic projection neurons originating from the VP (*Shea and Margoliash, 2003*). In sum, the VP appears an important

integration centre, where several pathways converge and perhaps form a closed loop system where dopaminergic midbrain nuclei can affect the basal ganglia to affect song output and vice versa, or cholinergic projections affect premotor cortical nuclei HVC and RA, based on error-signals originating from upstream auditory cortices (*Mandelblat-Cerf et al., 2014*). Our findings clearly complement functional studies by showing that the volume and most probably the organisation of fibres of passage becomes rearranged when birds achieve higher song copying accuracy levels.

MRI studies assessing brain-behaviour relationships along training programs to master complex motor skills have observed bi-directional neuroplastic changes (*Gryga et al., 2012*), that is, certain brain regions expand while others contract in response to training. Furthermore, depending on the training intensity and the timing of investigation, distinct parameter readouts can be obtained as also different neuroplastic mechanisms might be at play (*Sampaio-Baptista et al., 2014*). We speculate that continued improvements in song similarity as a form of vocal motor practicing might evolve towards an optimised and 'automatic performance' where redundant circuitries are pruned to facilitate optimal performance. In mammals, extended training leads to decreased number of task-activated neurons in the sensorimotor striatum, when proceeding from initial phases of novel skill learning to habitual performance of the task (*Ashby et al., 2010*). Also in birds, the vocal basal ganglia circuitry appears more important for initial song learning but functionally disengages with producing well-learned songs after song crystallisation (*Doupe et al., 2005*). The DTI metrics, on the other hand, provide an indirect estimate of the underlying tissue microstructure based on quantifying the average diffusion properties of water protons in a voxel. More specifically, FA quantifies the directional dependence of water diffusion (*Beaulieu, 2002*). As a result, alterations to FA are notoriously biologically unspecific as they can be caused by a wide variety of microstructural tissue reorganisations including altered axonal integrity, myelination, axon diameter and density, change in cellular morphology, etc. (*Beaulieu, 2002*; *Zatorre et al., 2012*; *Dyrby et al., 2018*). Moreover, the biological underpinnings responsible for the MRI readout are most probably always reflecting different processes happening in concert, in a coordinated way involving various different cell types. To unambiguously pinpoint the biological mechanisms responsible for the observed structural difference between good and bad learners, additional studies at the cellular and molecular level are required.

Our data-driven brain-wide analyses also pinpointed a fourth structure, which has not yet been investigated thoroughly in the context of zebra finch song learning, that is a cluster co-localised with the tFA. The tFA carries projections from the basorostral nucleus in the rostral forebrain to the lateral arcopallium and caudolateral nidopallium. More specifically, the basorostral nucleus sends somatosensory and auditory information to the lateral arcopallium (*Wild and Farabaugh, 1996*), which in turn projects to jaw premotor neurons and vocal and respiratory effectors (*Wild and Krützfeldt, 2012*). Also in humans, somatosensory information from facial skin and muscles of the vocal tract is vital for proper perception and production of speech (*Tremblay et al., 2003*; *Ito et al., 2009*). Alternatively, these somatosensory and auditory descending projections may serve to control beak movements that modulate gape size during singing, as gape size can affect the acoustic properties of individual syllables (*Hoese et al., 2000*). Taken together, even though this tract is not considered part of the traditional song control system, it might carry neuronal projections that are necessary for proper adjustment of vocalisations and be strengthened by vocal practicing. The precise role of the basorostral nucleus and the tFA in sensorimotor song learning needs to be further investigated in future studies that employ tools capable of dissection the observed relationship between brain structure and song copying accuracy level up to the mechanistic causal level.

In conclusion, the present findings clearly illustrate that as pupils produce more accurate copies of the tutor song, the structural properties of the CM, VP and tFA change. Most intriguingly, however, the final song performance relates to the microstructural tissue properties of the secondary auditory area NCM already at the early start of the sensory learning phase. Overall, our findings together with several parallel findings in humans, open new avenues in understanding brain-behaviour relationships related to speech acquisition and performance, both of which are imperative for successful communication, and again underscore the importance of early life rearing conditions and role models in defining future proficiency levels of complex multi-sensory learning processes.

# Materials and methods

**Key resources table**

| Reagent type (species) or resource | Designation | Source or reference | Identifiers | Additional information |
|---|---|---|---|---|
| Strain, strain background (*Taeniopygia guttata*, Male) | | Other | | Local breeding program |
| Chemical compound, drug | isoflurane | Abbott, Illinois, USA | | |
| Software, algorithm | Statistical Parametric Mapping | SPM | RRID:SCR_007037 | |
| | AMIRA | Amira | RRID:SCR_007353 | |
| | Matlab | Matlab | RRID:SCR_001622 | |
| | Sound Analysis Pro | SAP | RRID:SCR_016003 | |

## Animals and ethics statement

Male (n = 16) and female (n = 19) zebra finches (*Taeniopyiga guttata*; only males sing), bred in the local animal facility and were housed in individual cages together with an adult male (tutor), an adult female and one or two other juvenile zebra finches. At around 10 dph the juvenile birds were randomly assigned to an adult couple. This way some birds were co-housed with their biological parents, while others were raised by foster parents (see *Supplementary file 12*). Each cage was shielded from its neighbouring cages so that the juvenile birds could hear all other birds of the room (6–12 other tutors and many other juveniles), but could interact (visual and auditory) with only one adult male bird. Research has shown that the juvenile birds will prefer to copy the song of the adult male bird with whom they can interact with (*Eales, 1989*). The ambient room temperature and humidity was controlled, the light-dark cycle was kept constant at 12 h-12h, and food and water was available ad libitum at all times. In addition, from the initiation of the breeding program until the juvenile birds reached the age of 30 days post hatching (dph) egg food was provided as well. The Committee on Animal Care and Use at the University of Antwerp (Belgium) approved all experimental procedures (permit number 2012–43 and 2016–05) and all efforts were made to minimise animal suffering.

## Data statement

All data acquired and processed in this study are available online (DOI https://doi.org/10.5061/dryad.mkkwh70vj).

## Study design

We obtained MRI data of each bird during the sensory phase (20 and 30 dph), sensorimotor phase (40 and 65 dph), crystallisation phase (90 and 120 dph) and one last time point well beyond the critical period for song learning (200 dph; *Figure 1A*). Each imaging session, we collected a 3D anatomical scan and Diffusion Tensor Imaging (DTI) data to evaluate respectively gross neuro-anatomy (volumetric analyses) and alterations to white matter tracts or intrinsic tissue properties. Starting from the advanced sensorimotor phase (i.e. 65 dph), we recorded the songs sung by the juvenile males the first day after each imaging session.

## Song recordings and analyses

To quantitatively evaluate the progression of sensorimotor learning and song refinement in male birds, we analysed the first 20 (undirected) songs sung in the morning after 'lights on' in Sound Analysis Pro (SAP[*Tchernichovski et al., 2000*]; http://soundanalysispro.com/). The undirected songs sung by the juvenile and adult male zebra finches and tutors were recorded in custom-build sound attenuation chambers equipped with the automated song detection setup implemented in SAP. All song analyses were performed off-line and calls and introductory notes were omitted from all

analyses. First, the motif length (ms) of the first 20 songs sung during the morning (starting from the initiation of the photophase) was measured after which each individual motif was manually segmented into its different syllables based on sharp changes in amplitude and frequency. The latter measure was chosen to avert inconsistent determination of the syllable ending caused by more silent singing towards the last part of the syllable. Second, several acoustic features that reflect the spectro-temporal structure of individual syllables were quantified, that is (1) pitch-related measures that inform on the perceived tone of sounds (including pitch, mean frequency, peak frequency and goodness of pitch), (2) Wiener entropy that quantifies the tonality of sounds and is expressed on a logarithmic scale where white noise approaches '0' and pure tones are characterised by large, negative Wiener entropy values, (3) syllable and inter-syllable interval duration. Furthermore, to evaluate syllable feature variability over the different ages, the standard deviation, as an estimate for vocal variability (*Scharff and Nottebohm, 1991*), was defined for each acoustic property. Next, similarity to tutor song was measured between song motifs using an automated procedure in SAP that quantifies the acoustic similarity between two songs based on pitch, FM, AM, goodness of pitch and Wiener entropy (*Tchernichovski et al., 2000*). Song similarity was calculated using the default settings of SAP (asymmetric comparisons of mean values, minimum duration 10 ms, 10 × 10 comparisons), and % similarity was used for statistical testing. Further, according to the method conceptualised by Scharff and Nottebohm, motif sequence stereotypy was computed, based on visual assessment of sequence consistency and linearity (*Scharff and Nottebohm, 1991*). Sequence linearity reflects how consistent notes are ordered within the song motif by counting the different transition types of each syllable of the motif. Sequence consistency quantifies how often a particular syllable sequence occurs over different renditions of a specific motif. Song sequence stereotypy is defined as the average of sequence linearity and sequence consistency.

## MRI data acquisition

All MRI data were acquired on 7 T horizontal MR system (PharmaScan, 70/16 US, Bruker BioSpin GmbH, Germany) and a gradient insert (maximal strength: 400 mT/m; Bruker BioSpin, Germany), combined with a quadrature transmit volume coil, linear array receive coil designed for mice, following a previously described protocol (*Hamaide et al., 2017*). First, the zebra finches were anaesthetised with isoflurane (IsoFlo, Abbott, IL; induction: 2.0–2.5%; maintenance: 1.4–1.6%). While anaesthetised, the physiological condition of the birds was monitored closely by means of a pressure sensitive pad placed under the chest of the bird to detect the breathing rate, and a cloacal thermistor probe connected to a warm air feedback system to maintain the birds' body temperature within narrow physiological ranges (40.0 ± 0.2) ℃ (MR-compatible Small Animal Monitoring and Gating system, SA Instruments, Inc). Next, we collected DTI data using a four-shot spin echo (SE) echo planar imaging pulse sequence with the following parameters: TE 22 ms, TR 7000 ms, FOV (20 × 15) mm$^2$, acquisition matrix (105 × 79), in-plane resolution (0.19 × 0.19) mm$^2$, slice thickness 0.24 mm, b-value 670 s/mm$^2$, diffusion gradient duration (δ) 4 ms, diffusion gradient separation (Δ) 12 ms, 60 unique non-collinear diffusion gradient directions and 21 non-diffusion-weighted (b$_0$) volumes. The entire DTI protocol was repeated twice to increase the SNR (total DTI scanning duration: 72 min). The field-of-view included the telencephalon and diencephalon, which contain the auditory system and brain areas implicated in song control, the cerebellum and parts of the mesencephalon. Last, we collected a T$_2$-weigthed 3D Rapid Acquisition with Relaxation Enhancement (RARE) dataset with these settings: TE 55 ms, TR 2500 ms, RARE factor 8, FOV (18 × 16×10) mm$^3$, matrix (256 × 92×64) zero-filled to (256 × 228×142), spatial resolution (0.07 × 0.17×0.16) mm$^3$ zero-filled to (0.07 × 0.07×0.07) mm$^3$, scan duration 29 min. The FOV of the 3D scan encapsulated the entire zebra finch brain. The entire scanning protocol took no longer than 2.5 hr. All animals recovered uneventfully after discontinuation of the anaesthesia.

## MRI data processing

We processed both DTI and 3D RARE scans for voxel-based analyses following in house established protocols (*Hamaide et al., 2017*; *Hamaide et al., 2018a*; *Hamaide et al., 2018b*), and using the following software packages: Amira (v5.4.0, FEI; https://www.fei.com/software/amira-3d-for-life-sciences/), ANTs (Advanced Normalization tools; (*Avants et al., 2011*); http://stnava.github.io/ANTs/), FSL (FMRIB Software Library; (*Jenkinson et al., 2012*); https://fsl.fmrib.ox.ac.uk/fsl/fslwiki/FSL), and

SPM12 (Statistical Parametric Mapping, version r 6225, Wellcome Trust Centre for Neuroimaging, London, UK, http://www.fil.ion.ucl.ac.uk/spm/) equipped with the Diffusion II toolbox (https://sourceforge.net/projects/diffusion.spmtools.p/) and DARTEL tools (*Ashburner, 2007*).

## Deformation-based morphometry

First, we masked the individual 3D RARE scans (Amira 5.4.0) so that the datasets only include brain tissue. Second, we used the serial longitudinal registration (SLR) toolbox embedded in SPM12 to create one average 'within-subject' 3D dataset for each animal based on the individual masked 3D RARE scans acquired at the different ages (*Ashburner and Ridgway, 2012*). The SLR generated an average 'within-subject' 3D ('midpoint average') and jacobian determinant (j) maps. The latter maps are derived from the deformation field that contains the spatial transformation that characterises the warp between the midpoint average and each individual time point image, and encode for each voxel the relative volume at a specific age with respect to the midpoint average. Next, the midpoint averages of all animals were inputted in the 'buildtemplateparallel' function of the Advanced Normalization Tools (ANTs *Avants et al., 2011*) to create a between-subject population-based template. We used this initial 'between-subject' template to extract tissue probability maps reflecting mainly grey matter, white matter, or cerebrospinal fluid using the FMRIB Automated Segmentation Tool (FAST *Zhang et al., 2001*) embedded in FSL. The three probability maps created in this step were used as tissue class *priors* for segmenting all individual midpoint averages using the default settings of the (old)segment batch in SPM12. The resulting tissue segments of the midpoint averages were used to create a segment-based template in Diffeomorphic Anatomical Registration through Exponentiated Lie Algebra (DARTEL) (*Ashburner, 2007*). Next, the ANTs-based $T_2$-weighted between-subject template was warped via the 'DARTEL: existing template' batch to spatially match with the segment template and this template was used as anatomical reference space for all voxel-based analyses (referred to as 'population-based template'). The jacobian determinant maps were warped to the reference space using the flow fields produced by DARTEL (with modulation to preserve relative volume differences existing between different subjects). Lastly, the warped modulated jacobian determinant maps were log-transformed to convert exponential growth patterns to linear patterns (*Ashburner and Ridgway, 2015*) and smoothed using a Gaussian kernel with FWHM (0.14 $\times$ 0.14$\times$0.14) mm$^2$.

## Diffusion tensor imaging

The DTI data were pre-processed in the Diffusion II toolbox embedded in SPM12. First, we realigned the DTI volumes to correct for subject motion following a two-step procedure: an initial estimation based exclusively on the $b_0$ images was followed by a linear registration including all ($b_0$ and diffusion-weighted) volumes. Second, we co-registered the realigned DTI volumes to the individual 3D dataset acquired at the same age using normalised mutual information as objective function for inter-modal within-subject registration. In parallel, each individual masked 3D RARE dataset was bias corrected and spatially normalised to the population-based template using a 12-parameter affine global transformation followed by nonlinear deformations. These estimated spatial normalisation parameters were applied to the co-registered DTI volumes so that the DTI data spatially matched the population-based template space. During this writing step, the diffusion data were upsampled to an isotropic resolution of 0.19 μm. In parallel, the diffusion vectors were adapted to account for potential (linear) rotations incurred by the realignment, co-registration and normalisation procedures using the 'copy and reorient diffusion information' tool of the Diffusion II toolbox. Then, the diffusion tensor model was applied to the diffusion-weighted and $b_0$-data to estimate the diffusion tensor and Eigenvalues ($\lambda_1$, $\lambda_2$, $\lambda_3$). The Eigenvalues represent the principle axes of the radii of the 3D diffusion ellipsoid. Based on the Eigenvalues, the Fractional Anisotropy (FA) maps were computed. FA is scaled between 0 and 1; where 0 refers to isotropic and 1 anisotropic diffusion properties. Typically, one expects high FA values in white matter regions that contain many coherently organised myelinated fibre tracts. The FA maps of male zebra finches 200 dph were averaged together to create an average FA map using the image calculator of SPM (e.g. third and sixth panel in *Figure 1E*).

Several quality controls were performed throughout the data acquisition and processing procedures. Those included a visual inspection for ghosting, excessive movement, and spatial correspondence of registered images to the reference space (after co-registration and spatial normalisation

procedures). Last, the DTI parameter maps were smoothed using a Gaussian kernel with FWHM of $(0.38 \times 0.38 \times 0.38)$ mm$^3$.

## Statistical analyses: song

To analyse whether the song parameters change from 65 to 200 dph, we set up mixed-effect models including age as fixed effect, subject as random effect, subject*age as random slope and –only for the syllable level– syllable identity as random effect nested within subject. Furthermore, to determine whether song % similarity was dependent on the tutor by which the birds were raised, a mixed-effect model was performed with 'tutor identity' as fixed effect and 'bird identity' as random effect. We used the restricted maximum likelihood method to fit the data and assessed significance using F-tests with Kenward-Roger approximation. If a significant main effect could be observed for any of the song features examined, Tukey's HSD (Honest Significant Difference) *post hoc* tests were performed to situate when in time actual changes occur.

## Statistical analyses: MRI

Even though the MRI data were thoroughly checked at several stages in the pre-processing, an additional quality control was performed based on outlier detection. This analysis identified four outliers which, upon visual inspection of the datasets, appeared in two out of four to be driven by excessively large ventricles at the 20 dph time points and appeared normal thereafter. The other two animals showed an abnormal cerebellar folding patterns (leading to suboptimal subsequent image registration) and were therefore excluded from all analyses. In addition, one animal had one missing DTI scan, leaving 30 zebra finches (14 males and 16 females) for voxel-based statistical testing of interactions and main effects of DTI parameter maps. Furthermore, technical issues with the recordings of one tutor caused that we were not able to quantify song similarity to tutor song of two juvenile birds (one being the MRI-based outlier). Other technical issues lead to the loss of song recordings of two juvenile birds at 90 dph. This leaves 54 data points for voxel-based statistical correlation analyses (12 juveniles with 4 time points and 2 juveniles with 3 time points).

We have used a two-step approach to analyse the MRI datasets. First, instead of deciding where to look by manually drawing ROIs (of for example the song control and auditory nuclei), we used data-driven image analysis techniques that are capable of localising the specific brain sites where a brain-behaviour relationship exists. Therefore, we performed brain-wide voxel-based statistical analyses to identify (1) brain regions where sex differences in local tissue properties originate throughout the song learning process, (2) which brain sites mature between 20 and 200 dph and (3) which brain sites exhibit a significant relationship between performance (similarity) and the structural properties of the brain (DTI or local volume). Second, after establishing where in the brain these relationships exist, we aimed at better understanding the nature of the outcome of the voxel-based correlational analyses. Therefore, we extracted the average DTI or DBM parameter value for each cluster using 'cluster-based ROIs' and used those average parameter values to perform repeated-measures correlation analyses (rmcorr) and to create visual representations of the 'nature' of the correlation. The outcome of the DBM analysis was published in *Hamaide et al. (2018a)*, while the male data were used in this study to correlate with the song outcome of the same birds.

## Voxel-based repeated-measures ANOVA: analysis of interactions (age*sex) and main effects

All voxel-wise statistical tests were executed in SPM12. A repeated-measures ANOVA was performed on smoothed DTI parameter maps, including 'subject' as random factor, and 'sex' and 'age' as fixed factors. This design allowed for testing within-subject effects including interactions (age*sex) and main effects (age). Unless explicitly stated, only clusters that survived a family-wise error (FWE) correction thresholded at pFWE <0.05 combined with a minimal cluster size (kE) of at least 5 voxels for DTI analyses, were considered significant. All statistical maps are displayed overlaid onto the population-based template.

A similar analysis, performed on the smoothed modulated jacobian determinant maps, was published previously *Hamaide et al. (2018a)*. For a comprehensive overview of the results, we refer the reader to that paper.

## Voxel-based statistical correlation analyses between structural MRI and song parameters

To explore potential relationships between robust measures of song performance and the structural properties of the songbird brain, voxel-based multiple regressions were set up between the smoothed MRI parameter maps (log mwj and FA maps) and % similarity or sequence stereotypy. This protocol is based on *Hamaide et al. (2018b)*; *Orije et al. (2020)*. When searching for correlations between local tissue volume and the song features, total brain volume was added to the statistical design as additional covariate. All statistical analyses were performed on the entire brain (brain tissue within FOV) and no thresholds were applied on the smoothed MRI parameter maps. Unless explicitly stated, we used the following two criteria to assess the significance of a cluster: (1) clusters should contain at least 5 or 20 contiguous voxels for respectively DTI and 3D RARE analyses (number of contiguous voxels is represented by $k_E$) and (2) the 'peak voxel' (based on T values) of the cluster should survive a family-wise error (FWE) correction for multiple comparisons thresholded at $p_{FWE}$ <0.05 (*Roiser et al., 2016*). Only clusters where both criteria were satisfied were considered significant. These cluster sizes correspond to the following volumes: DTI: volume of cluster of 5 voxels is 0.04332 mm$^3$ and 3D RARE: volume of cluster of 20 voxels is 0.03808 mm$^3$. Based on Nixdorf-Bergweiler (1996, Journal of Comparative Neurology), these cluster-size thresholds (5 voxels for DTI; 20 voxels for 3D RARE) are small enough to detect differences even the smallest areas of the song control system nuclei. Furthermore, using the same acquisition protocols (identical voxel sizes), we have been able to detect structural differences between groups or over time in these small areas (DTI: *Hamaide et al., 2017*; DTI and 3D RARE: *Hamaide et al. (2018b)*; 3D RARE: *Hamaide et al., 2018a*). To overcome concerns that statistical correlation analyses was done on some of the measures coming from the same individuals, we reanalysed the data with a more stringent Sandwich Estimator (SwE) toolbox (http://www.nisox.org/Software/SwE/) approach and a classic ROI-based analysis as outlined in the 'response to reviewers' (p3-10). We could detect the same regions, but they would not be considered significant if we would apply the same selection criteria for assessing the significance of a cluster as outlined above (*Roiser et al., 2016*). As the purpose was exploratory and validated with subsequent cluster-based ROI correlation analysis, we preserved the initial outcome conform the analysis of our earlier studies (*Hamaide et al., 2018b*; *Orije et al., 2020*).

In contrast with song similarity, no supra-threshold voxels could be observed between any combination of smoothed MRI parameter maps and song sequence stereotypy. Clusters detected by the voxel-based multiple regression analysis and voxel-based repeated measures ANOVA were converted to ROIs (termed 'cluster-based ROIs', conversion at $p_{uncorrected}$ <0.001 $k_E \geq 40$ voxels) of which the mean DTI metrics or modulated log-transformed jacobian determinant were extracted for *post hoc* statistical testing. Extracting the previously identified clusters at a more liberal cluster makes them slightly larger, that is DTI: volume of cluster of 40 voxels is 0.34656 mm$^3$; 3D RARE: volume of cluster of 40 voxels is 0.07616 mm$^3$. This cluster-based ROI approach is identical to the methods used in our other studies for example (*Hamaide et al., 2018a*; *Hamaide et al., 2018b*; *Anckaerts et al., 2019*).

## ROI-based analyses

The voxel-wise multiple regression in SPM does not allow including a random effect for bird identity. Consequently, by inserting repeated-measures data we violate the assumption of independency of measures. To correct for this potential confound, we performed two additional tests on the cluster-based ROI data derived from the voxel based multiple regression analysis and voxel based analysis of the interaction (FA: age*sex). Additionally, we delineated song control nuclei: HVC, LMAN, RA and Area X, from which we extracted the mean DTI metrics to examine whether there are correlations to song similarity within the song control system. Firstly, we employed a repeated-measures correlation analysis (*Bakdash and Marusich, 2017*) in Rstudio (version 1.1.383, Rstudio, Boston, MA; http://www.rstudio.com/). This latter test informs on the existence of consistent within-subject correlations between the two variables. Hence, this analysis informs on potential song learning-related structural changes in the brain. Secondly, Spearman's ρ was calculated on data obtained at one specific age (65 dph, 200 dph) to test potential sources of between-bird variance in driving the correlations observed by the voxel-based analyses. An additional correlation analysis was run between the FA values at 20 dph and the song similarity at 200 dph, to determine whether the microstructural

tissue properties early in life already relate to song learning proficiency later in life. Spearmans' ρ was computed in JMP (Version 13, SAS Institute Inc, Cary, NC, 1989–2007).

To explore the possibility of predicting future song learning outcome (good or bad), we ran a mixed-effect model analyses on the cluster-based ROI parameters extracted from MRI data obtained at 20, 30 and 40 dph. Bird identity was added a random factor, and age (20, 30, 40 dph) and learning outcome (good, bad) were included as fixed effects in the model. The restricted Maximum Likelihood method was used to fit the data and significance was assessed using $F$-tests with the Kenward-Roger approximation.

All additional tests on the cluster-based ROIs were corrected for multiple comparisons using false discovery rate (FDR) based on the Benjamini-Hochberg procedure (*Benjamini and Hochberg, 1995*). In brief, this method ranks all p-values from smallest (rank $i = 1$) to largest (rank $i = n$). For each rank, the Benjamini-Hochberg critical value, $(i/m)Q$, is calculated. Where $i$ is the rank, $m$ is the total number of tests and $Q$ is the false discovery rate set at 0.05. All p-values smaller than and including the largest p-value where $p<(i/m)Q$ were considered significant. Overall, Benjamini-Hochberg is generally preferred method for multiple comparison correction since it controls for false positive discoveries, but also minimises false negatives (*Jafari and Ansari-Pour, 2019*).

The outcomes of all statistical tests performed in this study are summarised in the *Supplementary file 1*, *4–11*.

## Acknowledgements

We thank Dr S C Woolley and Dr D Vallentin for valuable discussions of the data and reading of earlier versions of the manuscript. The computational resources and services used in this work to build the population-based template were provided by the HPC core facility CalcUA of the Universiteit Antwerpen, the VSC (Flemish Supercomputer Center), funded by the Hercules Foundation and the Flemish Government – department EWI.

## Additional information

### Funding

| Funder | Grant reference number | Author |
|---|---|---|
| Fonds Wetenschappelijk Onderzoek | G030213N | Annemie Van der Linden |
| Fonds Wetenschappelijk Onderzoek | G044311N | Annemie Van der Linden |
| Fonds Wetenschappelijk Onderzoek | G037813N | Annemie Van der Linden |
| Fonds Wetenschappelijk Onderzoek | Hercules Foundation | Annemie Van der Linden |
| Fonds Wetenschappelijk Onderzoek | AUHA0012 | Annemie Van der Linden |
| Belgian Federal Science Policy Office | Interuniversity Attraction Poles P7/17 | Annemie Van der Linden |
| Fonds Wetenschappelijk Onderzoek | 1115215N | Jasmien Orije |

The funders had no role in study design, data collection and interpretation, or the decision to submit the work for publication.

### Author contributions

Julie Hamaide, Conceptualization, Formal analysis, Visualization, Methodology, Project administration; Kristina Lukacova, Jasmien Orije, Formal analysis; Georgios A Keliris, Supervision; Marleen Verhoye, Formal analysis, Supervision; Annemie Van der Linden, Conceptualization, Supervision, Funding acquisition

## Author ORCIDs

Jasmien Orije  https://orcid.org/0000-0002-6699-6221
Georgios A Keliris  https://orcid.org/0000-0001-6732-1261
Annemie Van der Linden  https://orcid.org/0000-0003-2941-6520

## Ethics

Animal experimentation: The Committee on Animal Care and Use at the University of Antwerp (Belgium) approved all experimental procedures (permit number 2012-43 and 2016-05) and all efforts were made to minimize animal suffering.

## Decision letter and Author response

Decision letter https://doi.org/10.7554/eLife.49941.sa1
Author response https://doi.org/10.7554/eLife.49941.sa2

## Additional files

### Supplementary files

• Supplementary file 1. Clusters displaying a time effect for FA in male and female zebra finch brains. FA stands for Fractional Anisotropy, one of the DTI metrics. The statistical maps were assessed at pFWE <0.05 and kE ≥5 voxels. 'Interaction age*sex' was assessed in a flexible factorial design and serves to unveil brain regions that display a sex-dependent trajectory during ontogeny 'Time effect' was assessed for each sex separately in a flexible factorial design and serves to unveil brain regions that change microstructurally over time. The 'Cluster level' and 'Peak level' columns refer to respectively the *p*-value (after 'Family Wise Error' correction for multiple comparisons) of the clusters and cluster extent (kE), and *p*- and *F*-values of the peak voxel of the clusters respectively, provided by SPM (see material and methods section).

• Supplementary file 2. Summary of the mixed-effect model analyses on the song scores. To test for a main effect of age, a mixed-effect model (n = 14–16; details in Methods section) was set up for each song score separately with age as fixed effect, subject as random effect, subject*age as random slope and –only for statistical analyses on the syllable feature level– syllable identity as random effect nested within subject. To test for a main effect of tutor, a mixed-effect model was executed for song similarity or sequence stereotypy separately with tutor as fixed effect and subject as random effect. The restricted Maximum Likelihood method was used to fit the data and significance was assessed using F-tests with the Kenward-Roger approximation.

• Supplementary file 3. Number of syllables in the songs of good and bad learners. '>65–68%' indicates that the birds always sung song copies with a song similarity to tutor song score of at least 68%, while '<65–68%' indicates that birds always sung songs with song similarity score lower than 65–68% similarity to tutor song. The 65–68% threshold was chosen arbitrarily based on the overall performance of the birds within the study.

• Supplementary file 4. Summary of the voxel-based multiple regressions (% similarity and FA). FA stands for Fractional Anisotropy, one of the DTI metrics. This table summarises the outcome of the voxel-based multiple regression based on 54 data points (12 birds with 4 time points and 2 birds with 3 time points). The 'Cluster' and 'Peak' columns refer to two different levels of assessing significance, respectively cluster-based inference and peak- or single voxel-based inference where the T- and p-value of the voxel with highest significance of the cluster is reported. Only clusters surviving pFWE <0.05 and kE > 5 voxels were considered significant. Grey font refers to clusters that were only visible when exploring the data at an exploratory statistical threshold, that is p uncorrected <0.001 and kE > 40 voxels. At this lower statistical threshold for visual assessment of the statistical maps, only clusters that still appeared significant with FWE correction at the cluster and peak level were considered relevant. The cluster extent for the clusters in the right tFA and right NCM are not reported as they could only be observed with an exploratory threshold (which influences cluster extent).

• Supplementary file 5. Summary of the voxel-based multiple regressions (% similarity and log mwj). 'log mwj' refers to the log-transformed, modulated and warped jacobian determinants. This table

summarises the outcome of the voxel-based multiple regression based on 54 data points (12 birds with 4 time points and 2 birds with 3 time points). The 'Cluster' and 'Peak' columns refer to two different levels of assessing significance, respectively cluster-based inference and peak- or single voxel-based inference where the T- and p-value of the voxel with highest significance of the cluster is reported. Only clusters surviving pFWE <0.05 and kE > 5 voxels were considered significant.

• Supplementary file 6. Benjamini-Hochberg FDR correction for multiple comparisons of rmcorr analyses. 'log mwj' refers to the log-transformed, modulated and warped jacobian determinants; FA stands for Fractional Anisotropy, one of the DTI metrics. rmcorr' is the repeated-measures correlation analysis. FDR rate = 0.05; number of tests = 8; i is the rank, m is the total number of tests and Q is the false discovery rate set at 0.05. Only those tests that survive FDR correction for multiple comparisons are highlighted in bold. '

• Supplementary file 7. Benjamini-Hochberg FDR correction for multiple comparisons of Spearmans' ρ analyses. 'log mwj' refers to the log-transformed, modulated and warped jacobian determinants; FA stands for Fractional Anisotropy, one of the DTI metrics. rmcorr' is the repeated-measures correlation analysis. FDR rate = 0.05; number of tests = 16; i is the rank, m is the total number of tests and Q is the false discovery rate set at 0.05. Only those tests that survive FDR correction for multiple comparisons are highlighted in bold.

• Supplementary file 8. Summary of mixed-effect model. This table summarises the results of the mixed-effect model analyses testing for an interaction between age (20-30-40 dph) and future good-bad learning outcome (n = 14 birds). The mixed-effect model includes a fixed effect for age and for learning outcome (good-bad), and a random factor for bird identity. The restricted Maximum Likelihood method was used to fit the data and significance was assessed using F-tests with the Kenward-Roger approximation.

• Supplementary file 9. Benjamini-Hochberg FDR correction for multiple comparisons of the interaction good-bad*age. FDR rate = 0.05; number of tests = 8; i is the rank, m is the total number of tests and Q is the false discovery rate set at 0.05; None of the tests survives FDR correction.

• Supplementary file 10. Benjamini-Hochberg FDR correction for multiple comparisons of the main effect good-bad. FDR rate = 0.05; number of tests = 8; i is the rank, m is the total number of tests and Q is the false discovery rate set at 0.05. Only those tests that survive FDR correction for multiple comparisons are highlighted bold.

• Supplementary file 11. Benjamini-Hochberg FDR correction for multiple comparisons of the spearman's correlation between % song similarity at 200 dph and FA at 20 dph. FDR rate = 0.05; number of tests = 5; i is the rank, m is the total number of tests and Q is the false discovery rate set at 0.05. Only those tests that survive FDR correction for multiple comparisons are highlighted bold.

• Supplementary file 12. Detailed information on bio-parents, foster-parents and tutors of the male zebra finches.

• Transparent reporting form

## Data availability

All figures are provided with the relevant source data. All data acquired and processed in this study are available online (DOI https://doi.org/10.5061/dryad.mkkwh70vj).

The following dataset was generated:

| Author(s) | Year | Dataset title | Dataset URL | Database and Identifier |
|---|---|---|---|---|
| Hamaide J, Orije J, Lukacova K, Keliris GA, Verhoye M, Van der Linden A | 2020 | Data from: In vivo assessment of the neural substrate linked with vocal imitation accuracy | http://dx.doi.org/10.5061/dryad.mkkwh70vj | Dryad Digital Repository, 10.5061/dryad.mkkwh70vj |

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
