## [Decision Letter]

**Acceptance summary:**

This is an intriguing paper that identifies novel brain areas involved in influencing the copying accuracy of birdsong. The paper identifies areas not previously linked to song learning, hence opening up new future investigations. Together with the extensive response to reviewers, this paper is an extensive resource for the birdsong community.

**Decision letter after peer review:**

Thank you for submitting your article "In vivo assessment of the neural substrate linked with vocal imitation accuracy" for consideration by *eLife*. Your article has been reviewed by three peer reviewers, and the evaluation has been overseen by a guest Reviewing Editor and Barbara Shinn-Cunningham as the Senior Editor. The reviewers have opted to remain anonymous.

The reviewers have discussed the reviews with one another and the Reviewing Editor has drafted this decision to help you prepare a revised submission.

Summary:

This study represents a very interesting new approach in identifying brain areas involved in accurate copying of tutor songs by male zebra finches. Using a whole-brain data-driven structural MRI approach, the authors identify a number of brain areas that either change alongside the improvement in copying as the birds mature, or structurally predict how well a bird will copy the tutor's song on an inter-individual level. Interestingly, none of these structures are in the traditionally-identified song control circuitry (although many are in areas related to auditory processing).

All three reviewers thought this was an interesting paper, but all three would like to see some clarifications of particular points and/or some re-analyses to drive home the message even more strongly. Below, I list the revisions that are required to improve the paper for publication in *eLife*.

Essential revisions:

Most of the essential revisions relate to the way the data analysis was performed and/or how the data were presented and discussed. No additional experiments are required.

1) A bit more detail on the birds and their experiences would be welcome. Clearly, some of the birds were exposed to the same tutor. But were some of the juveniles the offspring of the same parents? It would be helpful to know to what extent this information could be used to control for innate learning biases. Also, we are not requesting additional experiments, but it would have helped interpret the early left NCM FA values and learning outcomes if the juveniles had not been exposed to a tutor until after 30 days, to separate effects of the tutor song from innate properties of the birds.

2) MRI analysis 1 – Repeated measures: The authors used a two-step approach: first they ignore the fact that some of their measures were coming from the same individuals to perform their statistical analyses at the voxel level, and then, based on these results, used a ROI-based approach to take into account the repeated measures. We think that this approach is flawed because the selection of the voxels to determine the ROI is inaccurate (since each data point was considered to correspond to one subject). The result section should not present results where the repeated measure aspect of the dataset is not taken into account (first section of Results). We are aware that SPM does not currently allow analysing longitudinal datasets where the number of measures is not the same for all the subjects (as this is the case here). It seems that the authors have two options: (1) either they discard the 2 subjects for which they only have 3 data points and use a within-subject design for balanced designs in SPM; (2) or, even better, keeping their 14 subjects, they use the SwE toolbox (http://www.nisox.org/Software/SwE/) that seems to be able to handle unbalanced longitudinal datasets. Mean centering should allow the distinguishing between within- and between-subject effects (cf Guillaume, Hua et al., 2014, NeuroImage, 94).

3) MRI analysis 2 – explaining for non-experts: The manuscript falls short for a general audience in detailing how various structures were identified and assigned significance. One issue relates to the requirement for clusters > 40 contiguous voxels. What is the diameter of a sphere containing that many voxels? And does this volume threshold exclude smaller song nuclei, such as HVC, LMAN, DLM, or Avalanche? Finally, it would help a general reader to report the scale of FA, rather than just reporting absolute values.

4) MRI analysis 3 – correcting for multiple comparisons: The authors need to choose how they want to correct for multiple comparison in their voxel-based approach (voxel wise or cluster wise). If the authors choose a cluster-based approach, they should justify the first p value threshold used to obtain the clusters (recent published recommendations about how to choose these thresholds should be followed and mentioned). If they choose a voxel-size approach, they should justify their minimum cluster size.

5) MRI analysis 4 – Positive controls: A potentially noteworthy feature of the current study is that the only significant anatomical changes were detected in regions outside of the classical song system. But numerous studies have shown that the structure of various song control nuclei (HVC, RA, Area X) changes markedly over the period in which these measurements were made (increasing in volume between 20 and 60 days, and increasing in myelination between 20 and 100 days, eg). Further, some early structural changes in the song system (spine density and dynamics in HVC) are correlated with copying outcome. I would be more confident in the current results if the authors could show that their method is sensitive to structural changes within the song system that are known to occur during development, even (or perhaps especially) if these changes are not correlated with song learning outcomes.

6) Relating MRI to song learning – age: Please clarify how age is controlled for or used in the analysis. Do the authors just go from the assumption (based on the data, maybe) that copying accuracy increases with age, and that the two variables are therefore inextricably confounded? Or is there a way to separate maturation (age) effects from changes related to copying accuracy? It will also make it easier for the reader to understand statements like "However, individual improvements in song learning resulted in a lower local volume of the CM (left: p=0.0126; right: p=0.0075; Figure 3D)", which now may be difficult for some readers to assess, because age is not visible in the figure.

7) Relating MRI to song learning – representing changes in copying accuracy; This study includes multiple measures, using Fractional Anisotropy in Figure 2, and local volume in Figure 3. In both figures we see correlations between song similarity and MRI measure, but we do not see the time course of song learning. Therefore, statistical claims such as correlation between improvements in song learning and a lower local volume of the CM cannot be judged visually from the data as currently presented. To address this, authors should present figures, similar to Figures 2 and 3 but instead of showing the similarity vs MRI, present the similarity gains vs. MRI. For example: for each bird, you present similarity (day 90) – similarity (day 65) vs. MRI on day 90. This will allow the reader to judge visually (and not only statistically) if any of the MRI measure correlates with learning. In addition, it would be nice to have a figure illustrating this statement (e.g., showing similarity gains vs. right NCM activation): "Surprisingly, a small cluster in the right NCM displayed, in addition, a significant repeated-measures correlation."

8) Relating MRI to song learning – dichotomizing copying accuracy: Regarding the correlation between MRI properties at days 20/40dph and learning accuracy, the authors should justify why they dichotomised song accuracy (good vs. bad learners) rather than simply taking the% of song similarity. Why don't the authors test whether MRI properties at day 20 (or 30) allow predicting vocal learning accuracy at day 200 (expressed as% of song similarity)?

9) Discussion – mechanisms: The authors are careful to note the lack of explanatory power in these correlative measurements, which is good. But they need to say more about how they think these developmental changes might relate to learning. What are we supposed to make with the finding that the FA changes over development when there is no link to what that value represents in the songbird's brain? Going back to an earlier point, it would help to see that this method can detect FA changes related to increased myelination of the song system, which is dramatic and presumably should generate a large signal. That said, the authors need to discuss in depth how such changes (decreased volume, increased FA) could be related to better learning.

10) Discussion – Novelty: Prior studies have shown correlations between NCM functional properties and song learning outcome, between CM functional properties and vocal error detection, and between VP and song copying. A strong feature of the current study is that it provides independent validation that these regions correlate with song copying, but given the earlier work, the current findings are not wholly novel even if they are useful contributions. On the other hand, the tFA result is entirely novel but what this fiber tract is needs to be more fully described. The authors are quick to link it to the projections from the basorostral nucleus, but we are uncertain whether such a precise assignment can be made with these methods. Is this distinct from other fiber tracts in this general region, including parts of the occipitomesencephalic tract? Showing some conventional histology of the tFA in relation to the MRI data would be helpful here. And the VP finding is quite timely, given the recently emerging evidence of the role of this structure in song learning. Further, VP is the only region that showed significant correlations in both FA and volume with learning. Perhaps the manuscript should highlight the tFA and VP findings more strongly, while casting the NCM and CM data are more confirmatory in nature, to emphasize novelty.

[Editors' note: the decision after resubmission follows.]

Thank you for resubmitting your article "in vivo assessment of the neural substrate linked with vocal imitation accuracy" for consideration by *eLife*. Your revised article has been evaluated by a guest Reviewing Editor and Barbara Shinn-Cunningham as the Senior Editor.

We really appreciate the time you have put into writing long, thought-out responses to each of the reviewers' comments.

However, looking at the revised manuscript, it looks like very little has changed. The reasons the reviewers make these constructive comments is not so that you can explain things to them, but so that you can change the manuscript in such a way that readers with similar questions to the reviewers would find their questions already answered in the manuscript. Thank you for the changes you have already made.

I am therefore requesting that you please incorporate the responses you have made to the reviewers into the revised manuscript. Once I receive this revised manuscript, I will send it out for a second review with regards to the technical side of the MRI protocols and analysis methods.

I will here summarize which changes still need to be made to the manuscript itself:

1) Thank you for adding the table to the Supplementary materials. Could you please also add a few sentences to the Discussion laying out what your data can and cannot distinguish between, and what the obvious next studies would be to work out those distinctions? You have done this in the rebuttal, so it should not be difficult to add a bit to the Discussion.

2) This is probably the most important change. Since you have run the analysis now in a more appropriate manner, we feel that you should replace the original analysis with the new analysis, not just add the new analysis as an addendum to the paper. If the new analysis changes the outcomes of the study, then the Results and Discussion should be changed accordingly.

3) Thanks for what you've already added. However, I think you misunderstood the main question asked by the (non-MRI specialist) reviewers here. They just wanted to know how big those clusters were in real life, and how this compares to known song structures. You have clearly done all the calculations for the rebuttal. Now please incorporate that information also into the manuscript.

4) Does adding "peak voxel" to that sentence clarify the reviewers' question? I am not expert in this area, so cannot judge this. I will assume for now that it does.

5) Please do add the additional song control structure data to the manuscript. Other readers, thinking the same as the reviewers, will appreciate that the method can detect changes over time, as should be the case in the song system, but these do not correlate with copying accuracy. You may also want to refer to your 2018 paper when discussing these extra data.

6) Every reader is going to wonder whether the correlation between changes in MRI signal (FA, etc) and song copying accuracy is just a side-effect of both changing with time. So you have to address this in the analysis. If it is, as you say, purely a question of brain behaviour correlations, and age does not mediate this relationship, then show that. If it turns out that age is a major mediator, and removes the correlation between brain and behaviour, then please discuss why the correlation does exist in some brain areas and not in others, which also change with age.

7) This point is related to point 6: by losing time, we don't know whether the main reason for the correlations is that both change over time in a similar way, or that the copying accuracy actually explains the "noise" in the trend over time. So it would be good if the authors could think of some way that allows readers to understand the distinction between parallel trends over time and (not-age-related) correlations between brain and behaviour.

8) Please do add Figure REB8 to the manuscript, wherever you see fit, and add reference to it in the Results.

9) Thank you for a good explanation. Please add some of it to the Discussion, so all readers can benefit from this insight.

10) Can you add something about the Hamaide et al., 2017, paper and how you have done your best to identify the tract (and VP) as best as possible in the Discussion? I am happy for you to keep the emphasis as is on the three main points (if consistent with the new analyses, see (2) above).

a) Please add this justification to the manuscript (in a much shorter form, of course)

b) If this is really relatively uncommon, maybe add another half sentence about why Ashburner and Ridgeway recommend this.

Thank you for the rest of the changes you have already made.

[Editors' note: further revisions were suggested prior to acceptance, as described below.]

Firstly, I would like to apologize for how long this has taken. The holidays got in the way of finding and selecting an extra MRI-expert to give an opinion on the dispute between yourself and the initial MRI expert among the reviewers.

We have now received this second opinion, and it can be found below. The evaluation has been overseen by a guest Reviewing Editor and Barbara Shinn-Cunningham as the Senior Editor. The reviewers have opted to remain anonymous. The reviewers have discussed the reviews with one another and the Reviewing Editor has drafted this decision to help you prepare a revised submission.

This extra review refers specifically to point number 2 in the previous decision letter. I would like to emphasize that all the other points still stand and would need to be acted upon in order for the paper to be accepted. However, I believe that for most of them, this is not difficult. In addition, because the reviews and the response to reviewers will be published alongside the manuscript, I don't mind if you refer in the main text to the response to reviewers to save space. However, it is crucial that such references are in the main text, because many readers will not scroll down to the reviewers' comments and the responses to such.

As for point number 2, where we asked you to replace the original analysis with the new SWe based analysis, we have now asked an independent expert for their opinion. Their response is as follows:

Reviewer #4:

The authors rely on the correlation analysis method to identify unique brain regions (voxels) in the songbird brains, of which the FA, an integrated/vectorized readout of the diffusion-based MRI signal, varies to specific song learning behavior during development. Previous reviewers raise the concerns on the statistic validity to specify the unique brain regions, in particular, the "circular issue" to define the ROIs based on the selected voxels (beyond a significance threshold).

From the revised manuscript, similar brain regions were highlighted using the new analysis method, which is encouraging. However, the authors have to assign smaller voxel sizes to preserve individual voxels above the statistic threshold, which may break the compensatory/correction rules for multiple comparison problems. This issue has been well reported in the literature and is faced by neuroimaging researchers routinely. In most cases, it is due to the rather small sample size to present the population given a certain size of variability. In the animal MRI field, it is a known problem.

The authors do observe some reliably detected spatial patterns in the songbird brains (n=14). One of the challenges for the voxel-wise analysis is to precisely register the brains from individual subjects to the same template. The mismatch of voxels across subjects leads to pseudo-negative statistic estimates, but if a smoothing step (averaging voxels) is applied, it may reduce the potential FA value differences across different conditions. It is a dilemma.

Here, I suggested two tentative ways to deal with the problem:

An intriguing observation is the symmetric observation of the brain regions (left and right brain nuclei are identified and voxel counts are provided in tables). One possible way to deal with the statistical issue is to create a mirror image for each subject. Then, the authors can just focus on the one-side hemisphere to redo their analysis (hopefully with sufficient power).

The second way is to define the ROI based on the songbird anatomy, but not by the voxel-wise analysis results. If the atlas-ROI can show specific correlation features, it can serve as an alternative way to support the voxel-wise results.

Overall, I see that the main results are convincing and novel. The authors should apply the correct statistical analysis strategy to retrieve their major discoveries in a more convincing way.

---

## [Author Response]

Essential revisions:Most of the essential revisions relate to the way the data analysis was performed and/or how the data were presented and discussed. No additional experiments are required.1) A bit more detail on the birds and their experiences would be welcome. Clearly, some of the birds were exposed to the same tutor. But were some of the juveniles the offspring of the same parents?

Yes, some of the juveniles were the offspring of the same parents. We added an overview that indicates for each juvenile who were the biological and foster parents (if applicable), in the Materials and methods section as Supplementary file 12.

It would be helpful to know to what extent this information could be used to control for innate learning biases.

Based on the current dataset we cannot make conclusions about the implications of innate learning bias. Such tests require a carefully balanced/controlled study design where genetic brothers are raised in different conditions: (1) by its biological father (tutor = bio father), (2) by foster fathers (one foster father (tutor) per genetic brother).

Given that the behaviour of the tutor and social interactions between the tutor and the juvenile males have been shown to be important influencers of the juveniles’ song learning performance (Chen et al., PNAS), the rearing conditions and tutor exposure should be carefully controlled for. A yoked experimental design similar to what was used by Chen, et al., 2016, could help understand the effect of social interaction versus innate learning bias (auditory experience with limited visual and physical interactions).

Furthermore, a recent study has found that the interactions between juvenile males and their (foster) mother also have important effects on the juvenile males’ song maturation/performance (Carouso-Peck and Goldstein, 2019). Therefore, after the introduction of a tutor, the juvenile male birds should no longer be housed together with an adult female zebra finch (to avoid any potential social influences on vocal learning/performance due to social interactions with adult female zebra finches).

We added the following to the Discussion section:

“Importantly however, we should note that the study design employed in this study does not allow distinguishing between the implications of innate learning bias (innate properties of the pupil) and social enhancers of the tutor (social enhancers that promote learning in pupils). […] Lastly, delaying tutor exposure to after the first measurement (e.g. first (MRI) measure at 30 dph and introduction to tutor at 31 dph), can help differentiate between innate learning bias and social enhancement of vocal learning.”

Also, we are not requesting additional experiments, but it would have helped interpret the early left NCM FA values and learning outcomes if the juveniles had not been exposed to a tutor until after 30 days, to separate effects of the tutor song from innate properties of the birds.

Yes, we definitely agree! The goal of this study was to explore brain-behaviour relationships that arise along vocal learning. Therefore, we designed a longitudinal study where we raised 14 juvenile male zebra finches in close-to-normal rearing conditions (in small groups: adult male couple + 1 to 3 juveniles) and collected MRI data along with song recordings at distinct phases in the song learning process. When designing the study, we were not aware of social enhancers of vocal learning and therefore did not take this effect into account. Of course, we will take this finding and the reviewers’ suggestion into account when designing future studies!

More specifically, if we were to perform a more in-depth study relating to innate learning bias and social enhancers of vocal learning, we would definitely delay tutor exposure so as to investigate potential effects of innate learning biases (collect data before 30 dph) and potential changes in MRI readout elicited by exposure to the tutor at 30 dph (collect MRI data after 30 dph). Furthermore, we would include a higher number of birds divided over different social rearing conditions as briefly outlined above.

We added the following to the Discussion section:

“Lastly, delaying tutor exposure to after the first measurement (e.g. first (MRI) measure at 30 dph and introduction to tutor at 31 dph), can help differentiate between innate learning bias and social enhancement of vocal learning.”

2) MRI analysis 1 – Repeated measures: The authors used a two-step approach: first they ignore the fact that some of their measures were coming from the same individuals to perform their statistical analyses at the voxel level, and then, based on these results, used a ROI-based approach to take into account the repeated measures. We think that this approach is flawed because the selection of the voxels to determine the ROI is inaccurate (since each data point was considered to correspond to one subject). The result section should not present results where the repeated measure aspect of the dataset is not taken into account (first section of Results). We are aware that SPM does not currently allow analysing longitudinal datasets where the number of measures is not the same for all the subjects (as this is the case here). It seems that the authors have two options: (1) either they discard the 2 subjects for which they only have 3 data points and use a within-subject design for balanced designs in SPM; (2) or, even better, keeping their 14 subjects, they use the SwE toolbox (http://www.nisox.org/Software/SwE/) that seems to be able to handle unbalanced longitudinal datasets. Mean centering should allow the distinguishing between within- and between-subject effects (cf Guillaume, Hua et al., 2014, NeuroImage, 94).

We have re-analysed the data for both FA and log mwj using the SwE toolbox suggested by the reviewers. The outcome is shown in Author response image 1 and Author response image 2.

**Author response image 1. respfig1:** Voxel-based multiple regressions using SwE showing the positive correlation between% similarity and FA. The statistical parametric maps present the outcome of the voxel-based multiple regression testing for a correlation between song similarity and FA and are visualised at puncorr<0.001 and kE≥4 voxels, and overlaid on the population-based template and scaled according to the colour-code (T values) on the left of each statistical map. The crosshairs point to the tFA (**A**), the VP (**C**), NCM (**E**) all in the left hemisphere and CMM (**G**) in the right hemisphere. The extent and significance of these clusters is summarized in Author response table 1. Graphs **B**, **D**, **F** and **H** visualise the nature of the correlation between song similarity and FA where individual data points are colour-coded according to bird-identity (i.e. one colour = one bird). The average within-bird correlation is presented by the coloured lines, while the black dashed line indicates the overall association between song similarity and FA, disregarding bird-identity or age. ‘r’ is the repeated-measures correlation (rmcorr) coefficient. The * indicates a significant rmcorr correlation between FA and% similarity in the CMM (p=0.00287). Full summary of the overall and rmcorr correlation for all clusters is given in Author response table 2.

**Author response image 2. respfig2:** Voxel-based multiple regressions using SwE showing the negative correlation between% similarity and log mwj. The statistical parametric maps present the outcome of the voxel-based multiple regression testing for a correlation between song similarity and local tissue volume and are visualised at puncorr<0.001 and kE≥20 voxels, and overlaid on the population-based template and overlaid on the population-based template and scaled according to the colour-code (T values) on the left of each statistical map. The crosshairs point to the VP (**A**) or the CM in the left hemisphere (**C**). The extent and significance of these clusters is summarized in Author response table 1. Graphs **B** and **D** inform on the nature of the association between song similarity (%) and log-transformed modulated jacobian determinant (log mwj; a metric reflecting local tissue volume). More specifically, the individual data points of the graphs are colour-coded according to bird-identity (i.e. one colour = one bird). The average within-bird correlation is presented by the coloured lines, while the dashed black line indicates the overall association between song similarity and log mwj, disregarding bird-identity or age. ‘r’ is the repeated-measures correlation (rmcorr) coefficient. The * indicates a significant rmcorr correlation between logmwj and% similarity in the VP (p=0.0245) and in CM (left: p=0.0175). Full summary of the overall and rmcorr correlation for all clusters is given in Author response table 2.

**Author response table 1. resptable1:** Summary of the voxel-based multiple regressions using SwE (% similarity and FA).

Correlation between	Cluster	Hemisphere	Cluster	Peak
k_E_	*p_FWE_*	*T*	*p_uncorr_*
% similarity and FA	tFA	Left	22	0.675	3.63	<0.001
Right	7	0.559	3.79	<0.001
NCM	Left	11	0.635	3.74	<0.001
CMM	Right	8	0.587	4.10	<0.001
VP		4	0.933	3.16	0.001
% similarity and log mwj	VP		852	0.201	4.09	0.000
CM	Left	6265	0.925	2.49	0.000
Right	9372	0.024*	5.17	0.000

FA stands for Fractional Anisotropy, one of the DTI metrics. ‘log mwj’ refers to the log-transformed, modulated and warped jacobian determinants. This table summarises the outcome of the voxel-based multiple regression based on 54 data points (12 birds with 4 time points and 2 birds with 3 time points). The ‘Cluster’ and ‘Peak’ columns refer to two different levels of assessing significance, respectively cluster-based inference and peak- or single voxel-based inference where the T- and p-value of the voxel with highest significance of the cluster is reported. When applying the selection criteria of p_FWE_<0.05 and k_E_>5 voxels, only the correlation between% similarity and log mwj of the right CM can be considered significant.

**Author response table 2. resptable2:** Summary of the voxel-based multiple regressions using SwE (% similarity and FA or log mwj).

Correlation between	Cluster	Hemisphere	Between- subject correlation	Within-subject correlation
Spearman’s ρ	p value	rmcorr r	rmcorr p
**% similarity and FA**	**tFA**	Left	0.728	<0.0001	0.175	0.274
Right	0.656	<0.0001	0.326	0.0378
**NCM**	Left	0.635	<0.0001	0.184	0.248
**CMM**	Right	0.618	<0.0001	**0.454**	**0.0029***
**VP**		0.592	<0.0001	-0.041	0.799
**% similarity and log mwj**	**VP**		-0.583	<0.0001	**-0.399**	**0.0094***
**CM**	Left	-0.346	0.0103	**-0.393**	**0.0111***
Right	-0.327	0.0158	**-0.397**	**0.0101***

‘log mwj’ refers to the log-transformed, modulated and warped jacobian determinants; FA stands for Fractional Anisotropy, one of the DTI metrics. ‘r’ is the repeated-measures correlation coefficient of the within-subject correlation analyses. Spearmans’ ρ informs on potential correlations between the MRI parameters and song similarity at a specific time point between birds. Tests that survive Benjamini-Hochberg FDR correction for multiple comparisons are highlighted in bold Author response table 3.

**Author response table 3. resptable3:** Benjamini-Hochberg FDR correction for multiple comparisons of rmcorr analyses.

MRI parameter	Cluster-based ROI	Hemisphere	*p* value	rank	(*i/m)Q*
FA	CMM	R	**0.0029***	**1**	**0.0063**
Log mwj	CM	R	**0.0094***	**2**	**0.0125**
Log mwj	CM	L	**0.0101***	**3**	**0.0188**
Log mwj	VP		**0.0111***	**4**	**0.0250**
FA	tFA	R	0.0378	5	0.0313
FA	NCM	L	0.248	6	0.0375
FA	tFA	L	0.274	7	0.0438
FA	VP		0.799	8	0.0500

‘log mwj’ refers to the log-transformed, modulated and warped jacobian determinants; FA stands for Fractional Anisotropy, one of the DTI metrics. rmcorr’ is the repeated-measures correlation analysis. FDR rate = 0.05; number of tests = 8; *i* is the rank, *m* is the total number of tests and *Q* is the false discovery rate set at 0.05. Only those tests that survive FDR correction for multiple comparisons are highlighted **bold**. ‘rmcorr’ is the repeated-measures correlation analysis.

We understand the concern that some of measures are coming from the same individuals to perform statistical correlation analyses. Reanalysing the data with SwE as requested picked up the same regions. Though if we would apply the same selection criteria (Roiser et al., 2016) using only clusters that survived a family-wise error (FWE) correction thresholded at pFWE<0.05 combined with a minimal cluster size (kE) of at least 5 or 20 contiguous voxels for respectively DTI and 3D RARE analyses, only the correlation between song similarity and volume changes in right CM would be considered as significant.

The more stringent SwE analysis also influences the rmcorr correlations in some regions. Since we extracted the average FA value or log mwj of a cluster to calculate rmcorr correlations, these average values change as the extent of the clusters change. In the original analysis, VP for example was a large cluster (479 voxels) and showed a significant within subject correlation of r=0.496*. In the SwE analysis, the cluster size of the VP was much smaller (4 voxels), causing the within subject correlation to decrease to r = -0.041. However, the outcome of the within-subject correlation analyses between song similarity and local volume (log mwj) remain similar.

The voxel based analysis (1) ONLY had exploratory purposes to uncover very specific brain sites where this brain-behaviour relationship exists, (2) was used successfully in two other papers (Hamaide et al., 2018, Orije et al., 2020) and (3) most importantly, lead to subsequent rmcorr analysis approving the appointed regions to have the investigated correlation. Using SwE instead, we had to bend the criteria of significance and number of clusters as defined for MRI (Roiser et al., 2016) in order to continue to the discovered rmcorr outcomes. We also feel that adding the SwE analysis to the supplementary data would only complicate things for the reader, and dilute the attention from the main message. We do not want to make this into a methodological paper, comparing different statistical analyses. I hope we could convince you to keep the original analysis and its outcome in the current paper as well.

We added a reference to direct the readers to the ‘response to the reviewers’ for more information about the supplementary analyses we performed. These clarifications were added to the Materials and methods: Voxel-based statistical correlation analyses between structural MRI and song parameters section.

“To overcome concerns that statistical correlation analyses was done on some of the measures coming from the same individuals, we reanalysed the data with a more stringent Sandwich Estimator (SwE) toolbox (http://www.nisox.org/Software/SwE/) approach and a classic ROI-based analysis as outlined in the ‘response to reviewers’ (p3-10). We could detect the same regions, but they would not be considered significant if we would apply the same selection criteria for assessing the significance of a cluster as outlined above (Roiser et al., 2016). As the purpose was exploratory and validated with subsequent cluster-based ROI correlation analysis, we preserved the initial outcome conform the analysis of our earlier studies (Hamaide et al., 2018; Orije et al., 2020).”

3) MRI analysis 2 – explaining for non-experts: The manuscript falls short for a general audience in detailing how various structures were identified and assigned significance. One issue relates to the requirement for clusters > 40 contiguous voxels. What is the diameter of a sphere containing that many voxels? And does this volume threshold exclude smaller song nuclei, such as HVC, LMAN, DLM, or Avalanche? Finally, it would help a general reader to report the scale of FA, rather than just reporting absolute values.

We have used a two-step approach in this study:

First, we performed brain-wide voxel-based statistical analyses to identify which brain sites exhibit a significant relationship between performance (similarity) and structural architecture (DTI or local volume). Instead of deciding where to look by manually drawing ROIs of e.g. the song control nuclei and auditory areas, we use voxel-based statistical methods that are capable of uncovering very specific brain sites where this brain-behaviour relationship exists.

For all voxel-based analyses, we used very strict criteria to identify significant clusters (excerpt from the Materials and methods: Statistical analyses: Voxel-based statistical correlation analyses between structural MRI and song parameters section):

“Unless explicitly stated, we used the following two criteria to assess the significance of a cluster: (1) clusters should contain at least 5 or 20 contiguous voxels for respectively DTI and 3D RARE analyses (number of contiguous voxels is represented by kE) and (2) the ‘peak voxel’ (based on T values) of the cluster should survive a family-wise error (FWE) correction for multiple comparisons thresholded at pFWE<0.05 (Roiser et al., 2016). Only clusters where both criteria were satisfied were considered significant.”

For all voxel-based statistical analyses, we used a cut-off of 5 contiguous clusters for DTI and 20 voxels for volume analyses. These cluster sizes correspond to the following volumes (see calculations below):

– DTI: volume of cluster of 5 voxels: 0.04332 mm³

– Volume analysis from 3D RARE images: volume of cluster of 20 voxels: 0.03808 mm³

– We refer the reviewers to Figure 2 of Nixdorf-Bergweiler (1996). These graphs include the volumes of four song control nuclei in juvenile male (and female) zebra finches. Based on these graphs and on the voxel and cluster volumes calculated above, the voxel and cluster volumes are small enough to detect differences in the song control nuclei and auditory areas.

In our previously published studies we have been able to identify structural changes in the brains of adult zebra finches when testing for structural differences between males and females (in e.g. HVC and LMAN: Hamaide et al., 2017), upon targeted brain lesioning (in e.g. HVC, DLM: Hamaide, Lukacova et al., 2018 NeuroImage) and when assessing volumetric changes in ontogeny (in e.g. LMAN: Hamaide et al., 2018). We have now added the longitudinal DTI data in male and female zebrafinches in the current paper confirm that the method allows to pick up sexual differential changes during ontogeny (the SCS) (Figure 1B-E).

In the graphs by Nixdorf-Bergweiler, you can see that the volume of most of the song control nuclei reaches adult (>100 dph) sizes at around 60 dph, which is the youngest age at which we obtained song production data in this study. Therefore, this volume threshold does not exclude detecting structural differences in smaller song nuclei such as LMAN and DLM, compared to larger song control nuclei such as Area X.

Second, after establishing where in the brain these relationships exist, we aimed at better understanding the nature of the voxel-based statistical analyses, i.e. to overcome limitations of voxel-based statistical testing in SPM by using rmcorr, and to create a graph to visualise the ‘nature’ of the correlation. Therefore, we extracted the average DTI or DBM parameter value for each cluster. To this end, we created ‘cluster-based ROIs’. These cluster-based ROIs are defined based on the statistical parametric maps. Instead of thresholding the maps at pFWE<0.05 (†) (strict threshold to define whether a cluster is significant or not), we extracted the clusters at p _uncorrected_<0.001 (‡) kE≥40 voxels (which makes the previously identified cluster slightly larger). This is an approach identical to studies previously published by our lab e.g. Hamaide et al., 2018 (brain development study); Hamaide et al., 2018; Anckaerts et al., 2019.

The volumes of these clusters correspond to:

– 3D RARE: volume of cluster of 40 voxels: 0.07616 mm³

– DTI: volume of cluster of 40 voxels: 0.34656 mm³

Calculation:

– 3D RARE: voxel size at acquisition: (0.07x0.17x0.16) mm³

– volume of one voxel: 0.001904 mm³

volume of 20 voxels: 0.03808 mm³ †

volume of 40 voxels: 0.07616 mm³ ‡

– DTI: voxel size at acquisition: (0.19x0.19x0.24) mm³

volume of one voxel: 0,008664 mm³

volume of 5 voxels: 0.04332 mm³ †

volume of 40 voxels: 0.34656 mm³ ‡

The present MRI correlation study considers only male zebra finches of >64 dph. Average volumes deduced from the data of Nixdorf-Bergweiler:

· LMAN: 0.2 mm³

· HVC: 0.5 mm³

· RA: 0.27 mm³

· Area X: 1.5 mm³

We have added some clarifications to the Materials and methods: Statistical analysis MRI section.

“We have used a two-step approach to analyse the MRI datasets. First, instead of deciding where to look by manually drawing ROIs (of for example the song control and auditory nuclei), we used data-driven image analysis techniques that are capable of localising the specific brain sites where a brain-behaviour relationship exists. […] The outcome of the DBM analysis was published in Hamaide, De Groof et al. (2018), while the male data were used in this study to correlate with the song outcome of the same birds.”

We have added some clarifications to the Materials and methods: Voxel-based statistical correlation analyses section.

“These cluster sizes correspond to the following volumes: DTI: volume of cluster of 5 voxels is 0.04332 mm³ and 3D RARE: volume of cluster of 20 voxels is 0.03808 mm³. […] This cluster-based ROI approach is identical to the methods used in our other studies e.g. (Hamaide et al., 2018, Hamaide et al., 2018, Anckaerts et al., 2019).”

“Finally, it would help a general reader to report the scale of FA, rather than just reporting absolute values.”

We have added some clarifying sentences to the Materials and methods: MRI data processing: Diffusion Tensor Imaging section:

“FA is scaled between 0 and 1; where 0 refers to isotropic and 1 anisotropic diffusion properties. Typically, one expects high FA values in white matter regions that contain many coherently organised myelinated fibre tracts.”

4) MRI analysis 3 – correcting for multiple comparisons: The authors need to choose how they want to correct for multiple comparison in their voxel-based approach (voxel wise or cluster wise). If the authors choose a cluster-based approach, they should justify the first p value threshold used to obtain the clusters (recent published recommendations about how to choose these thresholds should be followed and mentioned). If they choose a voxel-size approach, they should justify their minimum cluster size.

For all voxel-based statistical analyses, we have based our threshold for assessing significance on an Editorial in NeuroImage Clinical (Roiser et al., 2016). More specifically, only clusters that survived a family-wise error correction for multiple comparisons of pFWE<0.05 (peak voxel value), and consisted of at least 5 (DTI) or 20 (volume) contiguous voxels, were considered significant.

These thresholds are similar to the thresholds we used in our previously published papers (of which we added the references to the manuscript text).

We have added the following clarification to the text in the Materials and methods: Statistical analyses: MRI: Voxel-based statistical correlation analyses between structural MRI and song parameters section:

“Unless explicitly stated, we used the following two criteria to assess the significance of a cluster: (1) clusters should contain at least 5 or 20 contiguous voxels for respectively DTI and 3D RARE analyses (number of contiguous voxels is represented by k_E_) and (2) the ‘peak voxel’ (based on T values) of the cluster should survive a family-wise error (FWE) correction for multiple comparisons thresholded at *p_FWE_*<0.05 (Roiser et al., 2016).”

Furthermore, each figure showing the result of voxel-based analyses has a supplementary file showing the exact pFWE values at cluster level and peak level which is encouraged by (Roiser et al., 2016). The subscript of the figure mentions the statistical threshold used.

5) MRI analysis 4 – Positive controls: A potentially noteworthy feature of the current study is that the only significant anatomical changes were detected in regions outside of the classical song system. But numerous studies have shown that the structure of various song control nuclei (HVC, RA, Area X) changes markedly over the period in which these measurements were made (increasing in volume between 20 and 60 days, and increasing in myelination between 20 and 100 days, eg). Further, some early structural changes in the song system (spine density and dynamics in HVC) are correlated with copying outcome.

The reviewers refer to a study by Roberts et al. (Nature 2010) who found that: “Spine dynamics were measured in the forebrain nucleus HVC, the proximal site where auditory information merges with an explicit song motor representation, immediately before and after juvenile finches first experienced tutor song. Higher levels of spine turnover prior to tutoring correlated with a greater capacity for subsequent song imitation. In juveniles with high levels of spine turnover, hearing a tutor song led to the rapid (~24h) stabilization, accumulation and enlargement of dendritic spines in HVC. Moreover, in vivo intracellular recordings made immediately before and after the first day of tutoring revealed robust enhancement of synaptic activity in HVC. These findings suggest behavioural learning results when instructive experience is able to rapidly stabilize and strengthen synapses on sensorimotor neurons important to the control of the learned behaviour.”

There is an important difference in study design and hypothesis between the study by Roberts et al. and our study. For these experiments, Roberts et al. temporarily deprived birds of a tutor and investigated before versus after tutoring exposure. In our experiments, the juvenile birds already sing advanced copies of the tutor song. We are investigating relationships between how well birds are copying the tutor song and the structural properties of the brain, while Roberts et al. are addressing the first stages of song learning.

Furthermore, we would like to point out that, even though our in vivoDTI protocol is very sensitive to detect changes in microstructural tissue properties, changes in spine turnover should affect the entire nucleus and to a large extent if we want to pick it up with our brain-wide in vivo imaging tools.

I would be more confident in the current results if the authors could show that their method is sensitive to structural changes within the song system that are known to occur during development, even (or perhaps especially) if these changes are not correlated with song learning outcomes.

1)Regarding 3D RARE data for detecting volumetric anatomical changes during development:

We have recently published a longitudinal study on brain development in zebra finches based on similar volumetric data (obtained at 20, 30, 40, 65, 90, 120 and 200 dph) as we present in this manuscript. In the paper (Hamaide et al., 2018), we are capable of detecting sex differences in local volume that arise between 20 and 200 dph in e.g. HVC, RA and NIf. Furthermore, considering brain development, we find clear patterns of volume increase and decrease over e.g. the different phases of song learning.

Please see Author response image 3 and its legend from the paper as well as the figure on anatomy from the supplementary figures of this paper (Author response image 4):

**Author response image 3. respfig3:** Relative volume differences between consecutive sub-phases of vocal learning in male and female zebra finches. The statistical maps highlight voxels where the modulated jacobian determinants are larger (blue: volume decrease from first to later phase) or smaller (red; expansion from first to later phase) at the sensory phase compared to the sensorimotor phase, sensorimotor compared to the crystallization phase, or around crystallization compared to 200 dph. The sensory phase includes data obtained at 20–30 dph, the sensorimotor phase 40–65 dph, and the crystallization phase 90–120 dph. The statistical maps are color-coded according to the scale on the right (T-values; T = 5.09 corresponds to pFWE<0.05, no cluster extent threshold). Author response image 4 presents anatomical labels on the slices underlying the statistical maps. The white arrow in the horizontal slices points to LMAN. Abbreviations: dph: days post hatching; Do: dorsal; Ve: ventral; Ro: rostral; Ca: caudal. (Hamaide et al., 2018).

**Author response image 4. respfig4:** Bird brain anatomy. **A** informs on the different subdivisions of the zebra finch brain projected on the population-based template. The colors refer to pallium, subpallium, thalamus and hypothalamus, midbrain, pons and medulla. The drawing on the right subdivides the telencephalon in its different sub-regions delineated by laminae, and cerebellum. **B** illustrates sagittal slices of the bird brain including schematic atlas drawings obtained from the zebra finch histological atlas browser (Oregon Health and Science University, Portland, OR 97239; http://www.zebrafinchatlas.org (Karten et al., 2013), and MR-images extracted from the population-based template. The numbers below the sagittal slices appoint the approximate (~) distance (mm) from the midline. Anatomical areas visible on the T2-weighted MRI slices are appointed by numbers, while the letters indicate regions that are only visible on the schematic atlas drawings. **C** provides an overview of (the approximate position of) anatomical regions defined in (**A**) on horizontal slices derived from the population-based template. The numbers below the sagittal slices correspond to the approximate position (in ‘mm’ from the midline). Legend: a: MMAN; b: Field L2b; c: Area X; d: HVC; e: RA; f: basorostral nucleus; g: lateral arcopallium; h: basal nucleus of Meynert and ventral pallidum; 1: posterior commissure; 2: Field L; 3: NCM; 4: thalamic zone; 5: TSM; 6: anterior commissure; 7: LMAN; 8: striatum including Area X; 9: FPL (lateral prosencephalic fascicle); 10: MLd; 11: TeO or ventral part of the optical lobe; 12: entopallium; 13: medial and lateral portion of the caudal mesopallium (respectively CMM and CML). Abbreviations: Do: dorsal; Ve: ventral; Ro; rostral: Ca: caudal; L: left; R: right.

We added citations of these prior studies showing sexual dimorphism and volume changes during ontogeny.

We have added some clarifications to the Results section:

“The 3D anatomical dataset enabled us to assess regional changes in brain volume that arise over time (brain development) or between male and female zebra finch brains (sex differences; these data have been published (Hamaide et al., 2017). […] The present data enables us to extend on the latter, as the present study includes DTI data obtained in juvenile zebra finches.”

We have added some clarifications to the Discussion section:.

“Firstly, a previous study by our lab investigated brain development in (juvenile) zebra finches (Hamaide et al., 2018). This study clearly shows that most changes in brain volume occur relatively early (before 65 dph), and that the changes affect large portions of the brains. Furthermore, the same study shows that relatively large, wide-spread brain areas decrease in volume from 65 dph to 200 dph (the same time frame as this study). The clusters detected in the current study are much smaller and may perhaps overlap with only a small fraction of these large clusters.”

We have added some clarifications to the Materials and methods section.

“The outcome of the DBM analysis was published in Hamaide et al., 2018, while the male data were used in this study to correlate with the song outcome of the same birds.”

2) Regarding DTI data for detecting ultrastructural changes during ontogeny:

We have been able to detect sex differences in the brains of adult zebra finches (Hamaide et al., 2017). Furthermore, the volumes of the song control nuclei reach adult size at around 60 dph in male zebra finches. In this study, the data used for voxel-based correlation analyses are obtained at 65 dph and older. Therefore, the size of the nuclei should not interfere with our ability to detect differences.

Furthermore, we now added a chapter to the Results section: “Longitudinal structural MRI changes in the brains of maturing male and female zebra finches.” This chapter shows a comprehensive overview of (1) which brain sites develop differences in local tissue microstructure between male and female zebra finch brains during the song learning process, and (2) the microstructural brain changes that characterise the first 200 days of post hatch life in both male and female zebra finch brains. The results are summarized in figure 1B-E. This to show that our method is able to detect well known biological differences between male and female zebra finches (Nixdorf-Bergweiler 1996) and their changes during ontogeny. These results were further discussed in the Discussion.

We added to the Results section.

*“*Longitudinal structural MRI changes in the brains of maturing male and female zebra finches:

We set up a longitudinal study where we repeatedly collected structural MRI data of the entire zebra finch brain (Figure 1A). […] In addition, several clusters identified by the (voxel-based) interaction between age and sex over time (Figure 1 B) were only found to be significantly changing during ontogeny in males (indicated by the white dotted boxes in Figure 1E).”

Concerning ‘even (or perhaps especially) if these changes in the SCS are not correlated with song learning outcomes’. This was actually a very nice suggestion and an extra control for our outcome. We investigated the correlation between% similarity and FA in main song control system regions: Area X, HVC, LMAN and RA. The outcome of this analysis and text has been added as an extra chapter in the Results section: “Song learning accuracy does not trace back to the song control system”.

6) Relating MRI to song learning – age: Please clarify how age is controlled for or used in the analysis. Do the authors just go from the assumption (based on the data, maybe) that copying accuracy increases with age, and that the two variables are therefore inextricably confounded? Or is there a way to separate maturation (age) effects from changes related to copying accuracy? It will also make it easier for the reader to understand statements like "However, individual improvements in song learning resulted in a lower local volume of the CM (left: p=0.0126; right: p=0.0075; Figure 3D)", which now may be difficult for some readers to assess, because age is not visible in the figure.

We do find a main effect of age, that is, we find that –on average– the pupils’ song similarity scores are significantly higher at 200 dph compared to 65 dph. For some birds, song similarity to tutor song does not increase over the different ages (visible in Figure 2A of the paper).

The goal of the current study was to identify brain-behaviour relationships. More specifically, we set out to investigate whether skilled performance (song learning accuracy deduced by song similarity) could be traced back to the structural properties of the brain. Based on the graphs, it is evident that not all birds follow the same trajectory, some birds learn faster or better compared to others. Therefore, we do not consider ‘age’ as a suitable reference or correcting factor and we chose to not take age into account in the statistical analyses of the MRI data. We think the repeated-measures correlation (rmcorr) is more powerful.

“However, individual improvements in song learning resulted in a lower local volume of the CM (left: p=0.0126 rmcorr=-0.391; right: p=0.0075 rmcorr=-0.416; Figure 4D).”

This statement is based on the outcome of the repeated-measures correlation (rmcorr) test. Age is not controlled in the rmcorr test, but bird-identity is taken into account (repeated measures). The outcome of the test indicates that –on average– when birds sing more accurate copies of the tutor song (within-bird comparisons!) they appear to have a smaller CM. In the Discussion, we speculate that continued improvements in song similarity as a form of vocal motor practicing might evolve towards an optimized and ‘automatic performance’ where redundant circuitries are pruned to facilitate optimal performance explaining the volume decrease in some of the involved structures. A similar explanation is used in the paper of (Ocklenburg et al., 2018) where they discovered asymmetry in tissue microstructure in the planum temporale (related to auditory speech processing) in human subjects, measured by diffusion MRI and explaining their readouts as more efficient information processing.

A detailed explanation is added to the Discussion section.

“It is generally known that in normal rearing conditions song learning accuracy improves with age. […] Based on these observations, we conclude that even though we cannot fully remove age-effects, we strongly believe that the current findings are mainly driven by correlations between performance levels and the structural characteristics of the brain rather than purely brain development effects.”

7) Relating MRI to song learning – representing changes in copying accuracy; This study includes multiple measures, using Fractional Anisotropy in Figure 2, and local volume in Figure 3. In both figures we see correlations between song similarity and MRI measure, but we do not see the time course of song learning. Therefore, statistical claims such as correlation between improvements in song learning and a lower local volume of the CM cannot be judged visually from the data as currently presented. To address this, authors should present figures, similar to Figures 2 and 3 but instead of showing the similarity vs MRI, present the similarity gains vs. MRI. For example: for each bird, you present similarity (day 90) – similarity (day 65) vs. MRI on day 90. This will allow the reader to judge visually (and not only statistically) if any of the MRI measure correlates with learning. In addition, it would be nice to have a figure illustrating this statement (e.g. E.g., showing similarity gains vs. right NCM activation): "Surprisingly, a small cluster in the right NCM displayed, in addition, a significant repeated-measures correlation."

The time course of song learning is presented in Figure 2B (song similarity scores relative to age).

The hypothesis of this study was focussed on finding brain-behaviour relationships. Therefore, we opted for correlation analyses where we were interested in between-bird variance (e.g.: Do birds that sing a better copy of the tutor song have a bigger HVC?) and within-bird changes (e.g.: If pupils progressively (between 65 and 200 dph) sing a better copy of the tutor song, does this improvement correlate –on average across the birds– with a change in microstructural tissue properties in a specific brain site?). These questions motivate our choice of visualising the data. Indeed, the current figures do not provide precise information about the age of the bird (Figure 2B demonstrates that% similarity increase on average with age), but the graphs of Figure 6 do provide valuable information about the overall song copying accuracy of the bird and consequently inform about the between-bird variation in learning accuracy (some birds produce on average a better copy of the tutor song than other birds, and this between-bird variance partly drives the correlation). On these graphs you can appreciate that birds which have high% similarity, demonstrate high% similarity at all ages, and present FA-data shifted to higher values: making the measure FA a surrogate measure for their ability to copy the song. This important piece of information would be lost when drawing figures based on similarity gains. Furthermore, when drawing figures based on similarity gains, we present figures which are not presenting our main message, and therefore could be misleading for the reader. Nevertheless, we made such figures (Author response image 5) to illustrate this.

**Author response image 5. respfig5:** Absence of significant Spearmans’ ρ correlation between similarity gain and FA of NCM, tFA and VP.

Similarity gain was calculated as the difference of song similarity% between baseline 65 dph and subsequent time points (90, 120 and 200 dph). However, the gain in song similarity does not correlate to FA values in any of the regions that correlate with similarity% . This is mostly because the gain in song similarity does not take the difference between subjects into account. At the baseline of our song analysis 65 dph, good singers already show a high similarity% to the tutor song. Therefore, the further gain in song similarity% is limited. The gain in song similarity is an intuitive measure for song learning, but requires to be measured early enough to monitor a real progression. However, the subsong that juvenile zebra finches produce is highly variable and difficult to analyse, which is why we started measuring song output at 65 dph.

8) Relating MRI to song learning – dichotomizing copying accuracy: Regarding the correlation between MRI properties at days 20/40dph and learning accuracy, the authors should justify why they dichotomised song accuracy (good vs. bad learners) rather than simply taking the% of song similarity. Why don't the authors test whether MRI properties at day 20 (or 30) allow predicting vocal learning accuracy at day 200 (expressed as% of song similarity)?

We display the correlation between (FA at 20 dph) and (% song similarity at 200 dph) in Figure 6—figure supplement 2 and this confirms what we postulated. Only a positive correlation between FA at 20dph and% song similarity at 200 dph was found in the left NCM.

We added this figure as Figure 6—figure supplement 2 and refer to it in the Results section:

“Furthermore, FA in left NCM at 20 dph was positively correlated (p=0.01, ρ=0.662) to the% song similarity at 200 dph (Figure 6—figure supplement 2).”

This result was specific to the left NCM as none of the other regions identified in the voxel-based correlation analysis showed a similar predictive relationship (Figure 6C, Figure 6—figure supplement 1 and 2).

We have added some clarifications to the Materials and methods section.

“An additional correlation analysis was run between the FA values at 20 dph and the song similarity at 200 dph, to determine whether the microstructural tissue properties early in life already relate to song learning proficiency later in life.”

9) Discussion – mechanisms: The authors are careful to note the lack of explanatory power in these correlative measurements, which is good. But they need to say more about how they think these developmental changes might relate to learning. What are we supposed to make with the finding that the FA changes over development when there is no link to what that value represents in the songbird's brain? Going back to an earlier point, it would help to see that this method can detect FA changes related to increased myelination of the song system, which is dramatic and presumably should generate a large signal. That said, the authors need to discuss in depth how such changes (decreased volume, increased FA) could be related to better learning.

In reply to this, I would like to quote from ‘Zatorre RJ, Fields RD, Johansen‐Berg H. Plasticity in gray and white: Neuroimaging changes in brain structure during learning. Nat Neurosci. 2012; 15(4): 528‐ 536’ which is in the reference list of our paper: “current neuroimaging techniques cannot directly inform us about the underlying cellular events mediating the observed effects. Moreover, phenomena visible via MRI are likely never the result of a single process happening independently, but probably involve multiple coordinated structural changes involving various cell types. Conversely, neuroimaging techniques offer certain advantages as they can be repeatedly performed in the same individual and provide whole-brain measures of brain structure and function”

The following Figure from this paper (Author response image 6) illustrates this.

**Author response image 6. respfig6:** Candidate cellular and molecular mechanisms of FA changes. (**a**) Cellular events underlying changes detected by MRI during learning include axon sprouting, dendritic branching and synaptogenesis, neurogenesis, changes in glial number and morphology, and angiogenesis in gray matter regions. (**b**) Changes in white matter include axon branching, packing density, axon diameter, fiber crossing, and the number of axons, myelination of unmyelinated axons, myelin thickness and morphology, changes in astrocyte morphology or number, and angiogenesis (from Zatorre RJ et al., Nat Neurosci. 2012).

In the current paper we could divide the outcome (4 regions) into 1) fiber structures (tFA) and 2) gray matter regions where the gray matter could still be a mixture of gray matter and some fiber structures (VP).

I can only repeat what Zatorre et al.wisely wrote in their review on this topic “Neuroimaging changes in brain structure during learning” that multiple factors can be involved and this is what the readers of our paper should grasp as a message but also that this unbiased data driven imaging approach uncovered 4 regions involved in song copying accuracy and pave the way to further investigation of these regions such as testing the impact of specific neuromodulations on relevant brain networks and on song learning. We do hope the reviewers appreciate this new, different, complementary and highly valuable study approach that directs songbird neuroscientist to new and different study targets.

In relation to the comment ‘Going back to an earlier point, it would help to see that this method can detect FA changes related to increased myelination of the song system, which is dramatic and presumably should generate a large signal’. Figure 1 showing statistical FA maps during development/upon ageing- clearly illustrates changes in white matter structures (definitely involving myelination but also other features) in the entire brain including those surrounding the SCN, their mutual connections (contributing to changes in various lamina) and further down (tOM). These maps at the same time help to observe the bigger picture of (all) the changes.

We have added some clarifications to the Results section.

“The DTI datasets allows to establish spatiotemporal maps that indicate when and where in the brain neuroplastic changes in tissue microstructure occur (Hamaide et al., 2016). In the current study, we focus on the Fractional Anisotropy (FA) outcome, a metric derived from DTI data. FA quantifies the directional dependence of water diffusion and hence indirectly reflects specific microstructural tissue characteristics (Beaulieu, 2002). Note that alterations in FA can be caused by a wide variety of microstructural tissue re-organisations including altered axonal integrity, myelination, axon diameter and density, change in cellular morphology, etc. (Beaulieu, 2002, Zatorre et al., 2012, Dyrby, et al., 2018).”

We have added some clarifications to the Discussion section.

“As a result, alterations to FA are notoriously biologically unspecific as they can be caused by a wide variety of microstructural tissue re-organisations including altered axonal integrity, myelination, axon diameter and density, change in cellular morphology, etc. (Beaulieu, 2002, Zatorre et al., 2012, Dyrby et al., 2018). Moreover, the biological underpinnings responsible for the MRI readout are most probably always reflecting different processes happening in concert, in a coordinated way involving various different cell types. To unambiguously pinpoint the biological mechanisms responsible for the observed structural difference between good and bad learners, additional studies at the cellular and molecular level are required.”

10) Discussion – Novelty: Prior studies have shown correlations between NCM functional properties and song learning outcome, between CM functional properties and vocal error detection, and between VP and song copying. A strong feature of the current study is that it provides independent validation that these regions correlate with song copying, but given the earlier work, the current findings are not wholly novel even if they are useful contributions. On the other hand, the tFA result is entirely novel but what this fiber tract is needs to be more fully described. The authors are quick to link it to the projections from the basorostral nucleus, but we are uncertain whether such a precise assignment can be made with these methods. Is this distinct from other fiber tracts in this general region, including parts of the occipitomesencephalic tract? Showing some conventional histology of the tFA in relation to the MRI data would be helpful here. And the VP finding is quite timely, given the recently emerging evidence of the role of this structure in song learning. Further, VP is the only region that showed significant correlations in both FA and volume with learning. Perhaps the manuscript should highlight the tFA and VP findings more strongly, while casting the NCM and CM data are more confirmatory in nature, to emphasize novelty.

The reviewers clearly mention “Prior studies have shown correlations between NCM functional properties and song learning outcome, between CM functional properties and vocal error detection, and between VP and song copying” while we clearly demonstrate in the current paper the link with structural properties of these regions! And also… that this structural property has a predictive value for copying accuracy! Both things ARE NEW! Follow up studies can now be planned in a very targeted manner to unravel what this structural property is, but also which connections ‘starting from’ or ‘projecting to’ these regions ‘under which early live (tutoring) circumstances’ contribute to the observed changes in this region. The current study includes a massive amount of work providing a solid foundation for many follow up studies with different methodological approaches.

The location of tFA in our MRI outcome matches perfectly with tFA assignation in the drawing atlas of Nixdorf-Bergweiler and Bischof, 2007. As an illustration (not a validation), we did an extra effort to calculate DTI Fiber tracking (tractography) by copying as ‘seed’ the location of the crosshair (i.e. the voxel of highest significance) from the statistical maps to the super resolution track density images of the adult zebra finch used in the publication of Hamaide et al, 2017. These tracts shown in Author response image 7 shed more light on the extent of the fibers passing the crosshair.

**Author response image 7. respfig7:** Result of voxel based multiple regression in tFA (**A**) used as a seed for exploratory tractography (**B**). The statistical map (**A**) presents the outcome of the voxel-based multiple regression testing for a correlation between song similarity and FA (n=14). The crosshairs point to the tFA in the left hemisphere (**A**). Tractography clarifies the tracts running through tFA (**B**) cluster found with the voxel-based multiple regression. Seed-based fiber tractography itself was performed on ex vivo super-resolution reconstruction DTI acquired in male zebra finch brain created for Hamaide et al., 2017. The seeds were positioned at the level of tFA to filter out the relevant tracts from the whole brain probabilistic track density imaging using MRtrix3. More details on the methods used to acquire and process the tractography data can be found in (Hamaide et al., 2017).

We have added some clarifications to the Results section.

“Using various atlases of the zebra finch brain (http://www.zebrafinchatlas.org/; (Nixdorf and Bischof, 2007, Poirier et al., 2008, Karten et al., 2013) and high resolution tract tracings within the zebra finch brain (Hamaide et al., 2017), we identified that these clusters co-localize with two secondary auditory areas, i.e. the caudomedial nidopallium (NCM) and caudal mesopallium (CM), with a white matter tract that connects the basorostral nucleus to the arcopallium (frontoarcopallial tract (tFA) (Wild and Farabaugh 1996)), and with an area at the base of the telencephalon termed the ventral pallidum (VP).”

Furthermore, we found an additional cluster midsagittal near the striatum and mesopallium, extending laterally and caudo-ventrally adjacent to the septomesencephalic tract (TSM; sub-peak next to the TSM in the left hemisphere: *p_FWE_*=0.002 *T*=6.38; Figure 3C). Based on this spatial pattern and in accordance with the Karten-Mitra zebra finch brain atlas (http://www.zebrafinchatlas.org/; (Karten et al., 2013)), we identified this area as the VP.

In the Discussion we already included –to our knowledge– all the literature there is on tFA and related tracts in songbirds. From this it appears that the discovery of its involvement in song copying accuracy was as new to us as for the songbird community. Besides Martin Wild, whose coordinates I can no longer retrieve as he is retired, there is no one to ask help. I can as for now only conclude that our findings clearly illustrate that as pupils produce more accurate copies of the tutor song, training and controlling the upper vocal organs i.e. above the syringe including tong, beak, facial muscles.… is also essential for song copying accuracy. The reviewer’s suggestion to bring this to the forefront is very uncomfortable as the paper aims at pointing out that this tract is important and therefore interesting to look into in future studies maybe in comparison with homologue or similar tracts and their function in the humans. Overall, the present findings (i) add a new dimension to previously published data as we provide clear evidence of relationships between performance levels and the structural properties of four specific areas, (and not the functional properties as was shown before) (ii) identify a novel not-yet-explored brain area (tFA) in the context of song learning which deserves in-depth investigation in future studies and (iii) uncover that future performance levels can be predicted based on the structural properties of a secondary auditory region at the earliest stages of song learning. We actually emphasized that we were able ‘to go back in time’ and to relate song learning accuracy levels obtained at 65 dph (and older) to the structural properties of specific brain regions of the same birds at 20 dph before actual vocal practicing starts.

[Editors' note: the decision after resubmission follows.]

As for point number 2, where we asked you to replace the original analysis with the new SWe based analysis, we have now asked an independent expert for their opinion. Their response is as follows:Reviewer #4:The authors rely on the correlation analysis method to identify unique brain regions (voxels) in the songbird brains, of which the FA, an integrated/vectorized readout of the diffusion-based MRI signal, varies to specific song learning behavior during development. Previous reviewers raise the concerns on the statistic validity to specify the unique brain regions, in particular, the "circular issue" to define the ROIs based on the selected voxels (beyond a significance threshold).From the revised manuscript, similar brain regions were highlighted using the new analysis method, which is encouraging. However, the authors have to assign smaller voxel sizes to preserve individual voxels above the statistic threshold, which may break the compensatory/correction rules for multiple comparison problems. This issue has been well reported in the literature and is faced by neuroimaging researchers routinely. In most cases, it is due to the rather small sample size to present the population given a certain size of variability. In the animal MRI field, it is a known problem.The authors do observe some reliably detected spatial patterns in the songbird brains (n=14). One of the challenges for the voxel-wise analysis is to precisely register the brains from individual subjects to the same template. The mismatch of voxels across subjects leads to pseudo-negative statistic estimates, but if a smoothing step (averaging voxels) is applied, it may reduce the potential FA value differences across different conditions. It is a dilemma.

We smoothed our data in plane 2 times the voxel size. This is also what we did in our other papers. Smoothing is often recommended, for compensating imperfect registration, but also to improve statistics. Smoothing renders the data more Gaussian distributed, improving the validity of the commonly used Gaussian random field theory thresholding approach, which is used in SPM (Smith and Kindlmann, 2013).

Here, I suggested two tentative ways to deal with the problem:An intriguing observation is the symmetric observation of the brain regions (left and right brain nuclei are identified and voxel counts are provided in tables). One possible way to deal with the statistical issue is to create a mirror image for each subject. Then, the authors can just focus on the one-side hemisphere to redo their analysis (hopefully with sufficient power).

We don’t think this is a good solution as the regions that were found clearly show asymmetry, e.g. left NCM allows predictions from FA, while right NCM does not; see Figure 6). Overall, hemispheric specialization has been demonstrated in zebra finches by us in previous fMRI studies (Poirier et al., 2009) and others (Phan and Vicario, 2010).

The second way is to define the ROI based on the songbird anatomy, but not by the voxel-wise analysis results. If the atlas-ROI can show specific correlation features, it can serve as an alternative way to support the voxel-wise results.

We agree that a good ROI based analysis is necessary to support the findings of a voxel based analysis, since the voxel-wise multiple regression in SPM does not allow including a random effect for bird identity. Consequently, by inserting repeated-measures data we violate the assumption of independency of measures. To correct for this potential confound, we performed additional tests on ROI based data. One can debate which ROI’s should be used for this analysis, atlas based ROI’s or cluster based ROI’s. In our prior comments to the reviewers we added a section to the paper where we performed a ROI based analysis on several song control nuclei to confirm that song learning accuracy does not trace back to the song control system. We will expand on this section adding a supplementary table with the full results of a ROI based analysis. In relation to this ROI based analysis, it is important to recognize the following facts: (1) There is no such ‘brain atlas’ of the zebra finch where the entire-brain is covered with detailed ROI’s in analogy with rodent brain atlases (e.g. different cortices, different subregions of the striatum, of the hippocampus etc.). This is probably because the songbird brain has been mainly studied as neural substrate for song learning and production. There are atlases which provide a list of annotations (including our MRI based atlas:(Poirier et al., 2008)) but as I mentioned they do not cover the entire brain and do not provide details/subdivisions. (2) It is exactly the exploratory unbiased voxel based analysis that allowed us to ‘discover’ additional regions beyond the well-known song related ROI’s, and this is the main purpose and benefit. One of these discovered regions (VP) or ventral pallidum, was brought up in very recent literature (Hisey et al., 2018). The other one (tFA) or fronto-arcopallial tract is hardly described in literature and not in relation to copying capacity and thus would have never been selected as ROI for further investigation if we would not had obtained the outcome of the exploratory voxel based approach!

Taking this into account we performed a ROI based analysis to confirm the results of the voxel based analysis. For this we transferred relevant ROI’s from the Zebra finch atlas (Poirier et al., 2008). Including regions from the song control system: Area X, HVC, LMAN, RA; and regions from the auditory system: Field L, NCM, CMM. In addition, we delineated tFA and VP, which came up significantly in the voxel based multiple regression analysis. The surroundings of Area X and RA were also delineated as they showed a significant change during ontogeny in male zebra finches. This to confirm whether a ROI based analysis would also pick up these regions as significant. These delineations were made based on the contrast of the average FA map created for figure 1E of the main article.

Since we are dealing with repeated measures, violating the assumption of independence, there are two different ways in which to approach the correlation analysis. Firstly, spearman’s ρ was calculated on averaged song similarity and FA for each subject. This is a common solution to resolve the issue of non-independence. This renders the overall association or between-subject correlation. Secondly, we analysed the repeated measures correlation, which takes for each bird the repeated-measures into account and can provide inference on the common association between brain structure and song similarity across the group of birds. Both types of correlation analysis were performed on the ROI based FA data. The results are summarized in Author response table 4.

**Author response table 4. resptable4:** Summary of the ROI-based between- and within-subject correlation analysis (% similarity and FA).

ROI	Hemisphere	Between- subject correlation	Within-subject correlation
Spearman’s ρ	p value	rmcorr r	rmcorr p
Song control system	Area X	Left	0.0286	0.9228	0.267	0.0911
Right	-0.1824	0.5325	0.162	0.313
HVC	Left	-0.4989	0.0694	-0.271	0.0871
Right	-0.0725	0.8054	0.0806	0.616
LMAN	Left	0.0989	0.7366	-0.135	0.4
Right	-0.0242	0.9346	-0.0114	0.944
RA	Left	-0.3231	0.2599	0.029	0.857
Right	0.0462	0.8755	0.138	0.391
Auditory system	Field L	Left	-0.0593	0.8403	0.108	0.501
Right	-0.2923	0.3105	0.0996	0.536
NCM	Left	**0.6967**	**0.0056**	0.109	0.499
Right	**0.5473**	**0.0428**	0.185	0.247
CM	Left	0.4813	0.0814	-0.123	0.443
Right	**0.622**	**0.0176**	0.283	0.0732
Other	VP	Left	**0.5956**	**0.0246**	0.136	0.397
Right	**0.6088**	**0.0209**	0.236	0.138
tFA	Left	**0.6396**	**0.0138**	0.259	0.102
Right	0.3451	0.2269	**0.343**	**0.0283**
Area X surr.	Left	-0.1341	0.6477	0.193	0.225
Right	-0.0637	0.8286	0.293	0.0633
RA surr.	Left	-0.156	0.5942	-0.191	0.232
Right	0.0198	0.9465	0.242	0.128

The FA averaged ROI based correlation analysis finds the same significant regions as the voxel based analysis. Picking up significant correlations in NCM (bilateral), right CM, left tFA. Right tFA does not present a significant between-subject correlation, but it is significant correlated within-subjects. Using the complete regions to calculate repeated measures correlations, does not give the same results. There is now no longer a significant rmcorr in right NCM, or VP. Meaning that our previous analysis could detect a small subpart of both NCM and VP that demonstrates a within-subject correlation to song similarity, whereas the analysis of the average data of the entire region could no longer pick up a repeated measures correlation. IEG expression patterns also don’t ‘fill’ the entire NCM (Terpstra et al., 2004), argumenting that NCM is indeed a large region with different sub-regions of which the particular functions have not yet been uncovered. Our exploratory voxel based analysis at least uncovered which part of NCM is engaged in song similarity. This is the entire point: our exploratory study exactly allowed us to go beyond these atlas shortcomings and discover what subregions are ‘recruited’ for producing a perfect copy of the tutor song!

FA stands for Fractional Anisotropy, one of the DTI metrics. This table summarizes the ROI-based between subject correlation, calculated using spearman’s ρ on song similarity and FA data averaged per subject (12 data points, 1 for each of the 12 birds). The within-subject correlation was calculated using repeated measures correlations based on 54 data points (12 birds with 4 time points and 2 birds with 3 time points). Significant correlations are highlighted **bold**. Abbreviations: LMAN: lateral magnocellular nucleus of the anterior; NCM: nidopallium; caudomedial nidopallium; CM: caudomedial mesopallium; tFA: fronto-arcopallial tract; VP: ventral pallidum; Area X surr.: Area X surroundings, RA surr.: RA surroundings.